EMBO
Molecular Medicine

# PPARG is central to the initiation and propagation of human angiomyolipoma, suggesting its potential as a therapeutic target

Oren Pleniceanu[1,2,3,4], Racheli Shukrun[1,2,4], Dorit Omer[1,2], Einav Vax[1,2,4], Itamar Kanter[5], Klaudyna Dziedzic[1,2,4], Naomi Pode-Shakked[1,2,4,†], Michal Mark-Daniei[1,2], Sara Pri-Chen[1,2], Yehudit Gnatek[1,2], Hadas Alfandary[4,6], Nira Varda-Bloom[3], Dekel D Bar-Lev[1,2], Naomi Bollag[7], Rachel Shtainfeld[7], Leah Armon[7], Achia Urbach[7], Tomer Kalisky[5], Arnon Nagler[3,4,†], Orit Harari-Steinberg[1,2], Jack L Arbiser[8,9] & Benjamin Dekel[1,2,4,*] (iD)

## Abstract

Angiomyolipoma (AML), the most common benign renal tumor, can result in severe morbidity from hemorrhage and renal failure. While mTORC1 activation is involved in its growth, mTORC1 inhibitors fail to eradicate AML, highlighting the need for new therapies. Moreover, the identity of the AML cell of origin is obscure. AML research, however, is hampered by the lack of *in vivo* models. Here, we establish a human AML-xenograft (Xn) model in mice, recapitulating AML at the histological and molecular levels. Microarray analysis demonstrated tumor growth *in vivo* to involve robust PPARG-pathway activation. Similarly, immunostaining revealed strong PPARG expression in human AML specimens. Accordingly, we demonstrate that while PPARG agonism accelerates AML growth, PPARG antagonism is inhibitory, strongly suppressing AML proliferation and tumor-initiating capacity, via a TGFB-mediated inhibition of PDGFB and CTGF. Finally, we show striking similarity between AML cell lines and mesenchymal stem cells (MSCs) in terms of antigen and gene expression and differentiation potential. Altogether, we establish the first *in vivo* human AML model, which provides evidence that AML may originate in a PPARG-activated renal MSC lineage that is skewed toward adipocytes and smooth muscle and away from osteoblasts, and uncover PPARG as a regulator of AML growth, which could serve as an attractive therapeutic target.

**Keywords** angiomyolipoma; cancer stem cells; mesenchymal stem cells; PPARG; tuberous sclerosis complex

**Subject Categories** Cancer; Stem Cells; Urogenital System

## Introduction

Angiomyolipoma (AML) is the most common benign renal tumor and is characterized by a unique histology, consisting of blood vessels, smooth muscle, adipose tissue, and epithelioid cells in varying proportions (Folpe & Kwiatkowski, 2010). AML can develop both sporadically and as part of tuberous sclerosis complex (TSC) (Folpe & Kwiatkowski, 2010), an autosomal dominant disease characterized by the development of tumors in various organs. TSC significantly impacts patients' lives, mainly due to brain and kidney lesions. While the former represent the main cause of morbidity (e.g., seizures), the latter are the major cause of mortality. Renal lesions in TSC are diverse and include both epithelial (e.g., renal cysts) and mesenchymal lesions (AML). AML can be fatal when complicated by massive hemorrhage or renal failure (Crino *et al*, 2006). Moreover, an aggressive variant, termed "epithelioid" AML, has been described and shown to possess metastatic potential (Konosu-Fukaya *et al*, 2014). TSC is thought to result from loss of function of hamartin or tuberin (encoded by *TSC1* and *TSC2*, respectively), two tumor suppressors, normally acting in a complex to inactivate mammalian target of rapamycin complex 1 (mTORC1).

1   Pediatric Stem Cell Research Institute, Edmond and Lily Safra Children's Hospital, Sheba Medical Center, Ramat Gan, Israel
2   Division of Pediatric Nephrology, Edmond and Lily Safra Children's Hospital, Sheba Medical Center, Ramat Gan, Israel
3   Division of Hematology and Cord Blood Bank, Sheba Medical Center, Ramat Gan, Israel
4   Sackler Faculty of Medicine, Tel Aviv University, Tel Aviv, Israel
5   Faculty of Engineering, Institute of Nanotechnology, Bar-Ilan University, Ramat Gan, Israel
6   Institute of Nephrology, Schneider Children's Medical Center of Israel, Petah Tikva, Israel
7   The Mina and Everard Goodman Faculty of Life Sciences, Bar-Ilan University, Ramat Gan, Israel
8   Department of Dermatology, Emory University School of Medicine, Atlanta, GA, USA
9   Winship Cancer Institute, Atlanta Veterans Administration Hospital, Atlanta, GA, USA
    *Corresponding author. Tel: +972 3 5302445; Fax: +972 3 5303637; E-mail: benjamin.dekel@gmail.com or binyamin.dekel@sheba.health.gov.il
    †Correction added on 3 April 2017 after first online publication: affiliation 4 has been added for NP-S; affiliations have been corrected from 1,2,3 to 3,4 for AN

TSC1/2 inactivation results in enhanced mTORC1 activity, leading to unrestrained cell growth and proliferation. Hence, it is currently believed that TSC-related tumors arise in part due to mTORC1 activation (Kenerson *et al*, 2002). However, clinical findings suggest that additional signaling pathways contribute to their development. For instance, clinical trials in TSC patients have demonstrated that mTORC1 inhibitors fail to eradicate AML (Bissler *et al*, 2008, 2013). The uncovering of these additional pathways, however, has been hampered by the absence of an *in vivo* model of AML. Interestingly, TSC1/2-deficient animals develop various renal tumors, including renal cysts and carcinomas (both characteristic of TSC) but not AML (Kobayashi *et al*, 1995, 1999, 2001; Liang *et al*, 2014), underscoring the need for an animal model of human AML. In addition, this further supports the notion that dysregulation of the TSC1/2-mTORC1 pathway cannot fully explain AML growth. Although AML was initially considered a hamartoma, it was later shown to be a clonal lesion and thus a true neoplasm (Kattar *et al*, 1999), prompting the search for its cell of origin. However, the exact identity of the AML cell of origin has not yet been uncovered. Based on histopathology, AML is thought to derive from a perivascular epithelioid cell (PEC), a cell type whose normal counterpart is currently unknown. AML is therefore a member of the PEComa group of tumors, defined by the World Health Organization as mesenchymal tumors composed of histologically and immunohistochemically distinctive PECs (Bonetti *et al*, 1994). Importantly, the pathological diagnosis of AML requires co-expression of melanocytic (e.g., HMB-45) and muscle-related markers [e.g., αSMA (α smooth muscle actin)] (Folpe & Kwiatkowski, 2010). Notably, AML can also arise in extra-renal sites [e.g., liver (Goodman & Ishak, 1984), heart (Shimizu *et al*, 1994), and skin (Fitzpatrick *et al*, 1990)], indicating that its cell of origin is present throughout the body. Mesenchymal stem cells (MSCs), once considered the stem cells of mesenchymal tissues (Caplan, 2005), are currently perceived as a subpopulation of pericytes, residing in virtually every tissue, including the kidney (Crisan *et al*, 2008). We previously isolated and characterized an MSC-like cell type in the mouse kidney interstitium harboring broad mesenchymal potential (Dekel *et al*, 2006). Due to the various sources from which MSCs can be obtained, as well as their considerable heterogeneity, minimal criteria for their definition have been formulated (Dominici *et al*, 2006). These include plastic adherence, a typical antigenic profile and multipotency. In this report, we used serial xenografting of AML cells to establish an *in vivo* model of human AML, which recapitulated the biology of the tumor at the histological, immunohistochemical, and molecular levels. In order to uncover the mechanisms involved in AML growth, we interrogated gene expression along xenograft (Xn) propagation. Microarray gene expression analysis revealed strong activation of peroxisome proliferator-activated receptor gamma (PPARG), a nuclear receptor and transcription regulator (Lehrke & Lazar, 2005) that is expressed in common epithelial tumors (e.g., breast and esophageal carcinoma) (Takahashi *et al*, 2006; Yuan *et al*, 2012). Immunohistochemical stainings (IHC) confirmed PPARG activation at the protein level both in AML-Xn and in primary human AML. Consequently, we show that PPARG inhibition significantly and specifically halts the *in vitro* growth of both sporadic and TSC-related AML cells and strongly limits their tumor-initiation capacity. We further demonstrate that PPARG inhibition leads to downregulation of the TGFB1 pathway, and specifically by inhibition of *PDGFB* and *CTGF*, two cardinal regulators of MSC/pericyte proliferation and function. Accordingly, we demonstrate that AML-Xn initiating cells fit within MSC criteria, thereby implicating a PPARG-activated resident renal MSC/pericyte lineage, which is therefore skewed toward the adipogenic and myogenic lineages, as the cell of origin of renal AML.

## Results

### Establishment of a human AML xenograft model in mice

In order to gain insight into the mechanisms underlying AML development, we first wished to establish an *in vivo* model of human renal AML. For this purpose, we used two cell lines derived from two renal AML patients: "UMB", derived from a TSC-related tumor and "SV7", derived from a sporadic tumor (Arbiser *et al*, 2001). We subcutaneously injected $10^6$ cells of each cell line into NOD/SCID mice. While UMB cells generated a slow-growing tumor approximately 4.5 months post-transplant, none of the mice injected with SV7 cells developed a tumor within 6 months, consistent with previous reports (Arbiser *et al*, 2001). Dissociation of the UMB-derived tumor into single cells and re-injection of $10^6$ cells into secondary and then tertiary recipients resulted in tumor propagation and Xn formation. Upon serial injections, the interval necessary for tumor growth became increasingly and significantly shorter. Injection of first-generation Xn (T1)-derived cells resulted in the formation of a palpable tumor within 50 days ± 5 days, whereas third-generation Xn (T3)-derived cells generated tumors within

---

**Figure 1.  Characterization of AML xenografts (Xn).**

A   Growth interval between sequential Xn generations from 1[st] (T1) to 4[th] (T4), shown as mean ± SD ($n \geq 3$); *$P < 0.05$; **$P < 0.01$ (one-way ANOVA with Bonferroni *post hoc* test). The exact *P*-values are specified in Appendix Table S5.
B   T1-Xn harbor mainly undifferentiated cells with rare lipid-distended cells (arrows). In contrast, T4-Xn show the classic triphasic AML histology, consisting of vessels (left panel, arrow) surrounded by a thick layer of perivascular epithelioid cell (PEC)-like cells. Peripherally, a lipoid area, composed of lipid-distended cells, is seen. Furthermore, T4-Xn harbor "myoid" areas (right panel, arrow), composed of immature myocytes. Scale bar: 100 μm.
C   Immunohistochemical staining (IHC) of T4-Xn for HLA, showing positive staining of the undifferentiated, lipoid, and myoid (left, middle, and right panels, respectively) compartments. Scale bar: 50 μm.
D   IHC of T4-Xn for HLA (left panel) and pericyte marker α-SMA (right panel). Shown is a vessel, in which α-SMA[+] pericytes are HLA[+] (arrows), whereas endothelial cells are of mouse origin (arrowheads). Scale bar: 100 μm.
E   Immunofluorescent staining (IF) of T4-Xn for HLA (green) and CD31 (red), demonstrating formation of human-derived vessels. Scale bar: 20 μm.
F   IHC of T1- and T4-Xn for the diagnostic AML markers HMB-45 and α-SMA. HMB-45 is expressed in both T1- (left panel) and T4-Xn (middle panel) and stains both blood vessels and adipocytes. Scale bar: 200 μm. In contrast, only T4-Xn express α-SMA (right panel). Scale bar: 20 μm.
G   IHC of T1- and T4-Xn for pS6, demonstrating expression in both T1- (left panel) and, to a greater extent, T4-Xn (middle panel). Scale bar: 50 μm. Note that within tumor vessels, only perivascular cells express pS6 (right panel). Scale bar: 20 μm.

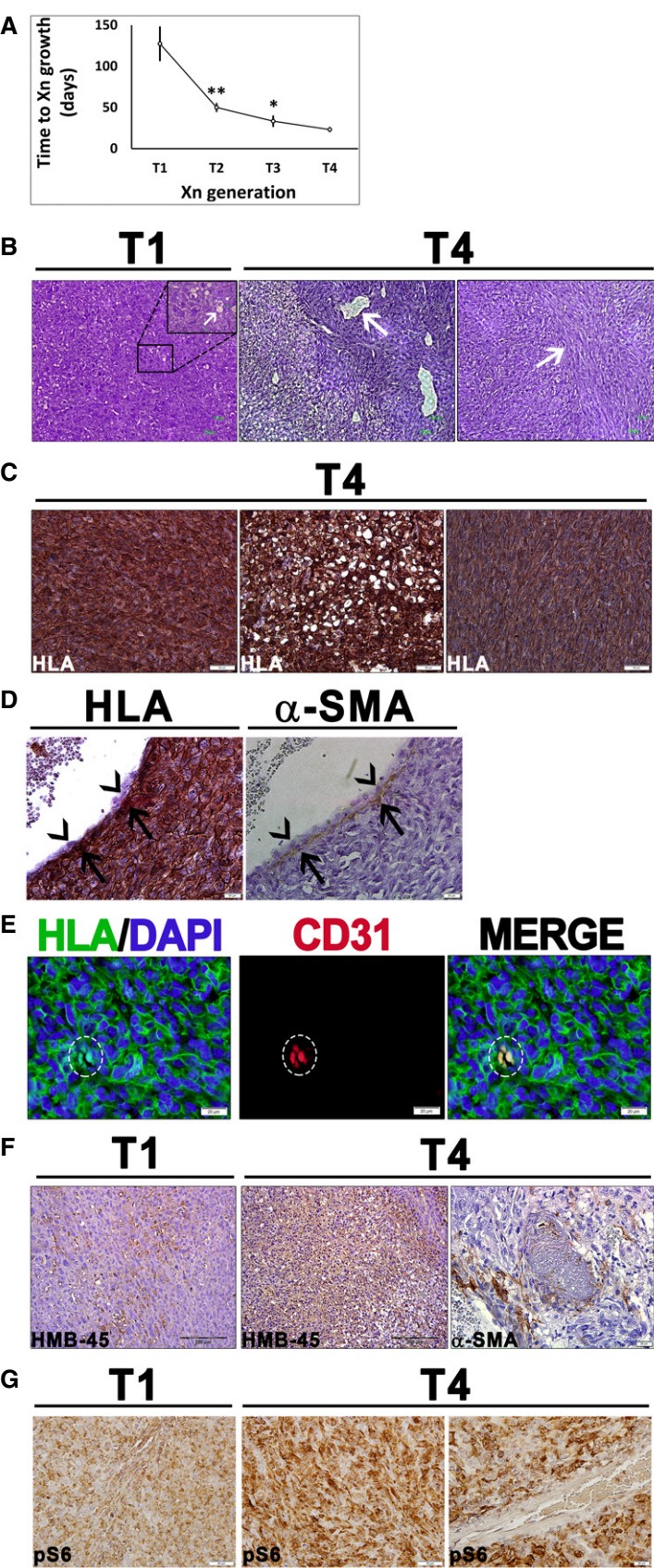

**Figure 1.**

23 days ± 3 days (Fig 1A). Upon histological analysis, T1-Xn consisted mainly of densely growing atypical cells (Fig 1B). Although the tumor did not exhibit characteristic AML histology, scattered lipid-containing cells could be noted (Fig 1B). Histological analysis of T4-Xn revealed the presence of the three cellular components of classical AML: (i) blood vessels surrounded by PEC-like cells; (ii) "lipoid" areas, composed of large masses of atypical adipocytes; (iii) "myoid" areas, composed of spindle-shaped myoid cells (Fig 1B). These three components were surrounded by dense masses of undifferentiated small hyperchromatic cells. Of note, the same histology was established in all repetitions carried out (*n* = 3), using independent UMB cells (Appendix Fig S1). In addition, the morphological appearance was specific to AML-Xn, and not detected in any other Xn model established in our laboratory (Appendix Fig S2). So as to ascertain that the various cellular phenotypes seen in the Xn result from differentiation of human AML cells, we carried out IHC staining of the Xn for the human-specific marker HLA. Indeed, the vast majority of the tumor stained positive for HLA, including the lipoid and myoid areas (Fig 1C). Interestingly, examination of vessels in the tumor revealed that only perivascular cells, but not endothelial cells, were HLA$^+$ (Fig 1D). IHC for the pericyte marker α-SMA confirmed that these HLA$^+$ cells surrounding vessels are indeed pericytes (Fig 1D). Accordingly, immunofluorescent staining (IF) of T1- and T4-Xn using HLA and human-specific CD31 antibodies revealed only rare tubular structures lined by HLA$^+$CD31$^+$ cells in T4-Xn (Fig 1E) and no such structures in T1-Xn. These results support the notion that vessel formation in AML-Xn involves human AML cells assuming the role of pericytes, but not endothelial differentiation of tumor cells. In contrast, several vessels in the tumor were completely mouse-derived, as manifested by negative HLA staining in both endothelial and perivascular cells (Appendix Fig S3). In order to determine whether the Xn represent genuine human AML tumors, we next queried whether they recapitulate AML at the immunophenotypical level. To this end, we stained T1- and T4-Xn for HMB-45 and α-SMA, two diagnostic AML markers (Folpe & Kwiatkowski, 2010). HMB-45 demonstrated strong positive staining in both Xn generations (Fig 1F). In contrast, α-SMA was expressed only in T4-Xn (Fig 1F), demonstrating positive staining in both perivascular and non-perivascular cells. Finally, to validate the identity of the AML-Xn, we carried out IHC of the Xn for pS6, a marker of mTORC1 activation, a key feature of human AML. Both T1- and T4-Xn demonstrated pS6 expression, with the latter exhibiting robust and diffuse staining for pS6 (Fig 1G). Thus, propagation by cell transfer was successfully carried out, establishing a transmissible source of bona fide human AML tissue for further experimentation. Taken together, the triphasic histology and the distinctive "melano-myocytic" phenotype establish the Xn as a valid *in vivo* model of human AML. The ability to derive these Xn from UMB cells strongly suggests that the latter represent an equivalent of the tumor cell of origin.

Notably, our results indicate that the characteristic vessels in AML do not result from endothelial differentiation of tumor cells. Rather, the latter seem to function as pericytes that recruit endothelial cells to form new vessels, in accordance with reports regarding the so-called PEC being the cell of origin of AML. In contrast, the other two lineages in AML (i.e., adipocytes and myocytes) seem to result from true differentiation of tumor cells.

### Molecular characterization of AML xenografts

We were next interested in using the Xn model to identify the molecular pathways driving tumor growth *in vivo*. Hence, we compared fifth-generation Xn ("T5") to first-generation Xn ("T1") via microarray-based global gene expression analysis. We first carried out Ingenuity Pathway Analysis (IPA) of significantly changed functions between T5- and T1-Xn cells (Fig 2A). This comparison disclosed decreased differentiation of connective tissue cells (consistent with enrichment for stem/progenitor cells of these lineages) and decreased phenotypes of normal MSCs lineages (e.g., diminished quantity of chondrocytes and less bone ossification) in T5- compared to T1-Xn. At the same time, there was an increase in colony formation and dysplasia (consistent with enrichment for tumor stem cells). In addition, we detected an increase in formation of endothelial cells and endothelial tube, as well as increased blood vessel rupture. We next explored the cellular processes accompanying AML growth by performing gene ontology (GO) enrichment analysis on T5- versus T1-Xn cells. Enriched biological processes and cellular components included cell division, anti-apoptosis, blood vessel development, fat cell differentiation, sarcomere, and melanosome (Table 1), all consistent with the cellular components of AML and the expression of melanocytic markers in this tumor. Concomitantly, we detected enrichment for binding of TGFB and PDGF, two cardinal players involved in the function and proliferation of pericytes/MSCs and in tumorigenesis (Furuhashi *et al*, 2004; Jian *et al*, 2006; Massague, 2008; Demoulin & Essaghir, 2014). We subsequently used IPA to pinpoint upstream regulators involved in AML-Xn propagation. Activated upstream regulators (Fig 2B) included mTOR, whose activation currently represents the main driving force behind TSC-related tumors, and PPARG, which was accompanied by strong upregulation of the *PPARG* transcript. Inhibited upstream regulators included TSC1 and TSC2, in accordance with AML pathogenesis. Detailed analysis of the mTOR pathway using IPA (Fig 2C) was consistent with known signaling in TSC. For instance, we noted activation of RPS6 and EIF4E, two downstream targets of mTORC1, which have been shown to be active in AML (Folpe & Kwiatkowski, 2010). In addition, the endothelial marker PECAM1 and the adipocytic marker FABP4, both indirect downstream targets of mTORC1, were upregulated, consistent with the cellular phenotypes seen in AML. Furthermore, the

---

**Figure 2. Microarray analysis of AML xenografts (Xn).**

A   Ingenuity Pathway Analysis (IPA) of increased (upper panel) and decreased (lower panel) functions in T5 (5$^{th}$ generation)- versus T1 (1$^{st}$ generation)-Xn.

B   Activated (upper panel) and inhibited (lower panel) upstream regulators in T5- versus T1-Xn according to IPA.

C   IPA of the mTOR pathway according to gene expression changes in T5- versus T1-Xn. Red/pink indicates increased gene expression, and green indicates decreased expression. *MTOR demonstrated activation as an upstream regulator with an activation *z*-score of 2.615 and a *P*-value of 1.93E-05.

D   Enriched Gene Ontology (GO) biological processes according to differentially expressed genes in T5-Xn compared to normal human kidneys.

**A**

| Increased |
|---|
| Colony formation |
| Dysplasia |
| Cell cycle progression of connective tissue cells |
| Formation of endothelial cells |
| Formation of endothelial tube |
| Rupture of blood vessels |

| Decreased |
|---|
| Differentiation of cells |
| Attachment of cells |
| Quantity of chondrocytes |
| Ossification of bone |
| Chemotaxis of BM cells |
| Aggregation of cells |

**B**

| Upstream regulator | Activation z-score | p-value |
|---|---|---|
| KRAS | 5.911 | 4.27E-15 |
| MYC | 4.96 | 3.85E-17 |
| MYCN | 4.292 | 2.23E-05 |
| PPARG | 2.714 | 9.86E-03 |
| MTOR | 2.615 | 1.93E-05 |
| Mek | 2.298 | 9.53E-03 |

| Upstream regulator | Activation z-score | p-value |
|---|---|---|
| TP53 | -3.158 | 1.26E-23 |
| TSC1 | -2.8 | 5.06E-02 |
| SP1 | -2.238 | 4.26E-07 |
| TSC2 | -2.045 | 4.51E-03 |

**C**

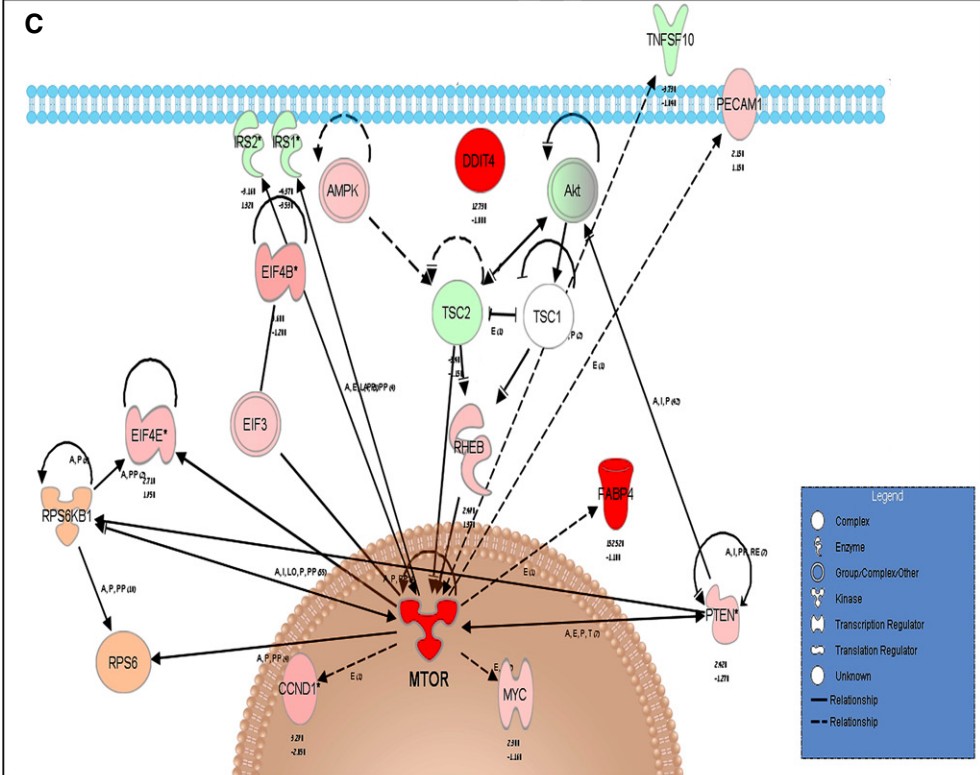

**D**

| Biological process | P-Value |
|---|---|
| Blood vessel morphogenesis (GO:0048514) | 6.02E-14 |
| Blood vessel development (GO:0001568) | 7.79E-14 |
| Regulation of cell motility (GO:2000145) | 6.18E-12 |
| Cell differentiation (GO:0030154) | 1.64E-11 |
| Angiogenesis (GO:0001525) | 1.75E-10 |
| Cell proliferation (GO:0008283) | 5.50E-08 |
| Cellular lipid metabolic process (GO:0044255) | 2.73E-05 |
| Fatty acid metabolic process (GO:0006631) | 9.52E-05 |
| Regulation of smooth muscle cell proliferation (GO:0048660) | 3.39E-04 |
| Muscle cell differentiation (GO:0042692) | 3.32E-02 |
| Stem cell differentiation (GO:0048863) | 3.57E-02 |

Figure 2.

Table 1.  Enriched GO (gene ontology) functions in T5-Xn compared to T1-Xn.

| Function | Enrichment score | Enrichment *P*-value |
|---|---|---|
| Cell division | 14.58 | 0.00 |
| Sprouting angiogenesis | 10.35 | 0.00 |
| Fat cell differentiation | 9.55 | 0.00 |
| Mitosis | 8.67 | 0.00 |
| Positive regulation of mesenchymal cell proliferation | 6.79 | 0.00 |
| Positive regulation of fatty acid oxidation | 6.17 | 0.00 |
| Anti-apoptosis | 5.36 | 0.00 |
| Melanosome | 8.05 | 0.00 |
| PDGF binding | 5.82 | 0.00 |
| Positive regulation of blood vessel endothelial cell migration | 5.15 | 0.01 |
| G1/S transition of mitotic cell cycle | 5.12 | 0.01 |
| Response to ER stress | 4.71 | 0.01 |
| Positive regulation of SMAD protein import into nucleus | 4.27 | 0.01 |
| Actin cytoskeleton reorganization | 4.24 | 0.01 |
| Positive regulation of fibroblast proliferation | 4.24 | 0.01 |
| Cholesterol biosynthetic process | 4.24 | 0.01 |
| Positive regulation of endothelial cell proliferation | 4.24 | 0.01 |
| TGFB binding | 4.71 | 0.01 |
| Myoblast fusion | 4.12 | 0.02 |
| Blood vessel development | 4.09 | 0.02 |
| Fibril organization | 3.83 | 0.02 |
| Positive regulation of PI3K activity | 3.83 | 0.02 |
| Sarcomere | 3.81 | 0.02 |
| Collagen binding | 4.09 | 0.02 |
| Actin filament-based movement | 3.66 | 0.03 |

analysis demonstrated compensatory inhibition of upstream regulators of mTORC1, such as AKT, IRS1, and IRS2, possibly reflecting a negative feedback loop that is also seen in AML (Folpe & Kwiatkowski, 2010). Inhibited upstream regulators included TSC1 and TSC2, in accordance with AML pathogenesis. Of note, alongside PPARG activation, we detected strong downregulation (5.4-fold decrease) of *WNT5A*. Interestingly, PPARG and WNT5A have been identified as master regulators of MSC/pericyte differentiation, promoting adipogenic and osteogenic differentiation, respectively (Xu *et al*, 2016). To further explore the relevance and reliability of the established AML-Xn model from a molecular perspective, we carried our bioinformatic analysis, comparing T5-Xn to normal human adult kidney tissue (AK), representing the AML tissue of origin. For this purpose, the normal borders of two human kidneys derived from nephrectomies were used. In accordance with the above results, we

detected strong and significant upregulation of *PPARG* (over 21-fold). Next, we applied GO enrichment analysis of genes showing fold change of ≥ 3 in expression between T5-Xn and AK. We detected enrichment of several key biological processes characterizing AML. These include angiogenesis, blood vessel development and morphogenesis, regulation of smooth muscle cell proliferation, muscle cell differentiation, cellular lipid metabolic process, cell proliferation, and cell differentiation (Fig 2D). Hence, the Xn model exhibits all the classical molecular features usually present in human AML tumors. Taken together, these results demonstrate that the Xn model mimics human AML at the molecular level, displaying, among others, strong activation of the mTOR pathway. As such, this model can be reliably used to study AML biology. Importantly, these findings suggest that the unique phenotype of AML results from a transcriptional program supporting vasculogenesis and differentiation into the adipogenic lineage, rather than other mesenchymal lineages (i.e., osteogenic and chondrogenic). This type of reciprocal differentiation pattern, which is highly characteristic of pericytes/MSCs, possibly implicates the latter in AML pathogenesis.

**PPARG activation accompanies AML-Xn propagation**

In an attempt to uncover novel therapeutic targets for AML, we decided to focus on genes which demonstrated strong activation along Xn propagation both at the transcript level and as upstream regulators. The gene that best fitted within these criteria was *PPARG*, which demonstrated a 42.69-fold increase at the transcript level and an activation z-score of 2.714 as an upstream regulator when comparing T5- to T1-Xn. Accordingly, IHC demonstrated significant upregulation of the PPARG protein in T4- compared to T1-Xn. Whereas T1-Xn was almost devoid of PPARG expression (Fig 3A), T4-Xn demonstrated strong PPARG staining. Of note, PPARG expression was not limited to adipocytes, but rather noted in all cellular components of the tumor, including the undifferentiated component, where strong nuclear staining could be seen. The same results were obtained in all repetitions carried out (*n* = 3) (Appendix Fig S1). Accordingly, IF for PPARG in T4-Xn cells demonstrated expression of PPARG, in the both cytoplasmic and nuclear compartments, highlighting its functionality in AML-Xn cells (Fig 3B). In order to gain insights into the possible role of PPARG in AML propagation, we performed network analysis of significantly changed (fold change ≥ 1.5) PPARG-dependent genes in T5- compared to T1-Xn (Appendix Table S1). Significant changes were noted in genes contributing to the phenotype of AML, including upregulation of regulators of angiogenesis (e.g., *VEGFA*) and adipogenesis (e.g., *FABP4* and *FABP5*). In addition, changes in gene expression consistent with tumor growth could be seen (e.g., upregulation of *CCND1* and *KRAS* and downregulation of *TP53* and *TNFSF10*). Furthermore, many PPARG-dependent genes involved in the mTORC1 pathway exhibited a significant change, consistent with mTORC1 activation, including activation of *PTEN* and downregulation of *AKT1/2* and *IRS1/2*. Taken together, these results indicate that PPARG is strongly activated during AML-Xn propagation. Importantly, the significant activation of PPARG as an upstream regulator, as reflected by the widespread changes in PPARG-dependent genes, indicates that *PPARG* is not only highly expressed along AML propagation, but also functions

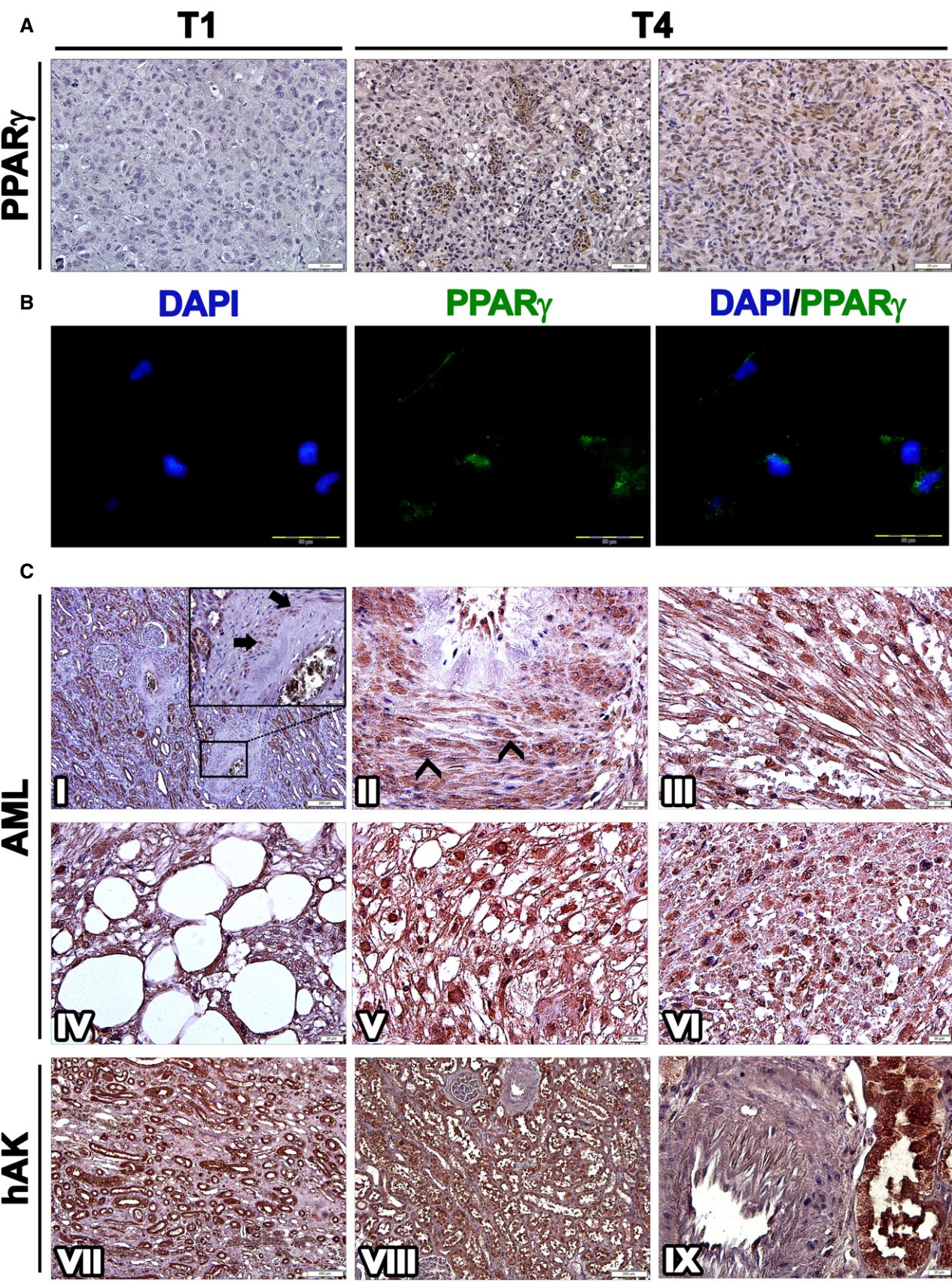

**Figure 3.**

◀

**Figure 3.   PPARG is strongly expressed in human AML.**

A   Immunohistochemical staining (IHC) of T1 (1$^{st}$ generation)- and T4 (4$^{th}$ generation)-xenografts (Xn) for PPARG. T1-Xn (left panel) demonstrate low expression of PPARG. In contrast, T4-Xn are strongly positive for PPARG. Abundant expression is seen both in adipocytic cells (middle panel) and in the undifferentiated non-adipocytic compartment of the tumor (right panel), where strong nuclear staining can be noted. Scale bar: 50 μm.

B   Immunofluorescent staining of T4-Xn cells for PPARG (green), demonstrating both cytoplasmic and nuclear expression. Scale bar: 100 μm.

C   IHC of primary AML tumors for PPARG. Normal renal tissue of the patient (I) demonstrates strong PPARG expression in tubules and endothelial cells and lack of expression in glomeruli. In vessel walls (magnified in I), PPARG expression can be noted in a small portion of pericytes (arrows). Within tumor tissue, numerous pericytes strongly expressing PPARG are seen (II), demonstrating both nuclear (arrowheads) and cytoplasmic expression. PPARG is also strongly expressed in the myoid and adipocytic compartments of the tumor (III and IV, respectively). Particularly strong nuclear expression is seen in epithelioid cells (V). Scattered within the tumor are small undifferentiated cells (VI), which are also positive for PPARG. Normal control human adult kidneys (hAK) demonstrate PPARG expression in all tubule segments, including collecting ducts (VII and VIII) but not in glomeruli. In contrast to normal and tumor tissue from AML patients, blood vessels in normal hAK do not harbor PPARG-expressing pericytes (IX). Scale bars: II–VI, IX: 20 μm; I, VII, VIII: 200 μm.

to regulate various other cellular factors and pathways. Of note, the robust expression of PPARG in both adipogenic and non-adipogenic tumor compartments strongly suggests that its role in AML is not limited to adipogenesis, but may be an oncogenic driver for other cell types.

## PPARG is strongly expressed in human AML

Having shown the strong activation of PPARG during AML-Xn propagation, we were interested in determining whether it is also expressed in primary human AML tumors. For this purpose, we performed IHC for PPARG in six sporadic AMLs, one TSC-related AML, and normal renal tissue from the same patients. Among these, one tumor was an extra-renal AML and three were fat poor, including one that demonstrated atypical epithelioid morphology (Appendix Table S2). Although the tumors were variable in terms of histology, PPARG was strongly expressed in all seven tumors, and in all histologic lineages of pericytes, myocytes, and epithelioid cells. Normal kidney sections from the healthy margins of these tumors showed strong PPARG expression in renal tubules and endothelial cells while glomeruli and the renal interstitium were negative for PPARG. Around blood vessels, scattered pericytes showing mainly cytoplasmic PPARG expression were noted (Fig 3C). In contrast, within AML tissue, blood vessels contained numerous pericytes strongly expressing PPARG, localized to both the nucleus and cytoplasm. Adipocytes, myocytes, and epithelioid cells were all strongly positive for PPARG, the latter two demonstrating prominent nuclear expression. Interestingly, within the tumor we detected small undifferentiated cells that also expressed PPARG. Finally, we analyzed PPARG expression in normal renal tissue derived from the margins of RCC specimens. Similar to normal kidney sections of AML patients, we detected PPARG expression in renal tubules and vascular endothelium, but not in glomeruli. However, pericytes in normal kidney tissue of non-AML patients were devoid of PPARG expression (Fig 3C). Taken together, these results indicate that human AML tumors strongly express PPARG, irrespective of their specific histology and cellular composition.

## Human AML cells are PPARG dependent and sensitive to PPARG antagonism

Having established that PPARG is strongly activated both during AML-Xn propagation and in primary AML, we were interested in examining the effects of PPARG inhibition on AML cells. To this end, we used GW9662, a specific PPARG inhibitor (Leesnitzer *et al*, 2002). We first incubated UMB and SV7 cells with varying concentrations of

GW9662 or vehicle and quantified cell numbers at 48 and 96 h, using the MTS assay, comparing, at each of the two time points, cells treated with a given concentration to control cells. GW9662-treated UMB cells exhibited significantly reduced cell growth compared to control, vehicle-treated cells, at concentrations as low as 30 μM, demonstrating a dose- and time-dependent response (Fig 4A). Similarly, GW9662 inhibited the growth of SV7 cells (Fig 4A) in a dose- and time-dependent manner, resulting in significantly lower proliferation at concentrations as low as 20 μM. Of note, these concentrations are similar or lower than previously reported for growth inhibition of other tumor types (Schaefer *et al*, 2005; Takahashi *et al*, 2006). We next examined the effect of PPARG inhibition on Xn cells, representing an enriched population of tumor-propagating cells. GW9662 significantly inhibited the growth of T4-Xn cells compared to control, vehicle-treated cells, demonstrating a dose- and time-dependent effect (Fig 4A). Interestingly, T4-Xn cells demonstrated an inhibitory concentration 50 (IC$_{50}$) of 32.2 μM, significantly lower than UMB cells, which showed an IC$_{50}$ of 36.0 μM (Appendix Fig S4), supporting the increased dependence of Xn cells upon PPARG activity. Accordingly, T5-Xn cells demonstrated an even lower IC$_{50}$, of 20.6 μM (Appendix Fig S4). In addition to reduced proliferation, UMB and T4-Xn cells treated with GW9662 showed a significantly altered phenotype, with loss of normal cellular morphology, compared to control, vehicle-treated cells (Fig 4B). Using Western blot analysis, we detected lower levels of PPARG protein in both UMB and SV7 cells following GW9662 treatment (Appendix Fig S5). In order to verify the effect of PPARG inhibition on AML cells, we applied another specific PPARG inhibitor, T007 (Lee *et al*, 2002), to UMB, SV7, and T5-Xn cells. Consistent with previous results, PPARG inhibition using this antagonist resulted in significant growth inhibition in all three cell types (Fig EV1). Subsequently, we carried out *PPARG* knockdown (KD) in UMB and SV7 cells using *PPARG*-shRNA lentiviral vectors. UMB and SV7 cells were infected with *PPARG*-targeting shRNA or control-shRNA, both consisting of a GFP-expressing cassette. *PPARG*-shRNA-infected cells demonstrated significant reductions in PPARG protein levels compared to control cells (Fig 4C). Next, in order to assess the effect of *PPARG* KD on cell proliferation, both cell lines were seeded in equal amounts (300,000 cells) of control and treated cells and recounted after 7 days. Consistent with the above results, *PPARG* KD resulted in a significant reduction in cell numbers (Fig 4D and E). In addition, we attempted to carry out *PPARG* knockout (KO) using CRISPR-CAS9 technology via specific *PPARG*-targeting sgRNA in both UMB and SV7 cells. Interestingly, following many repetitions in both cell types (*n* = 14), while all established clones demonstrated heterozygous *PPARG* KO (14/14), as validated by Sanger sequencing, we were unable to detect

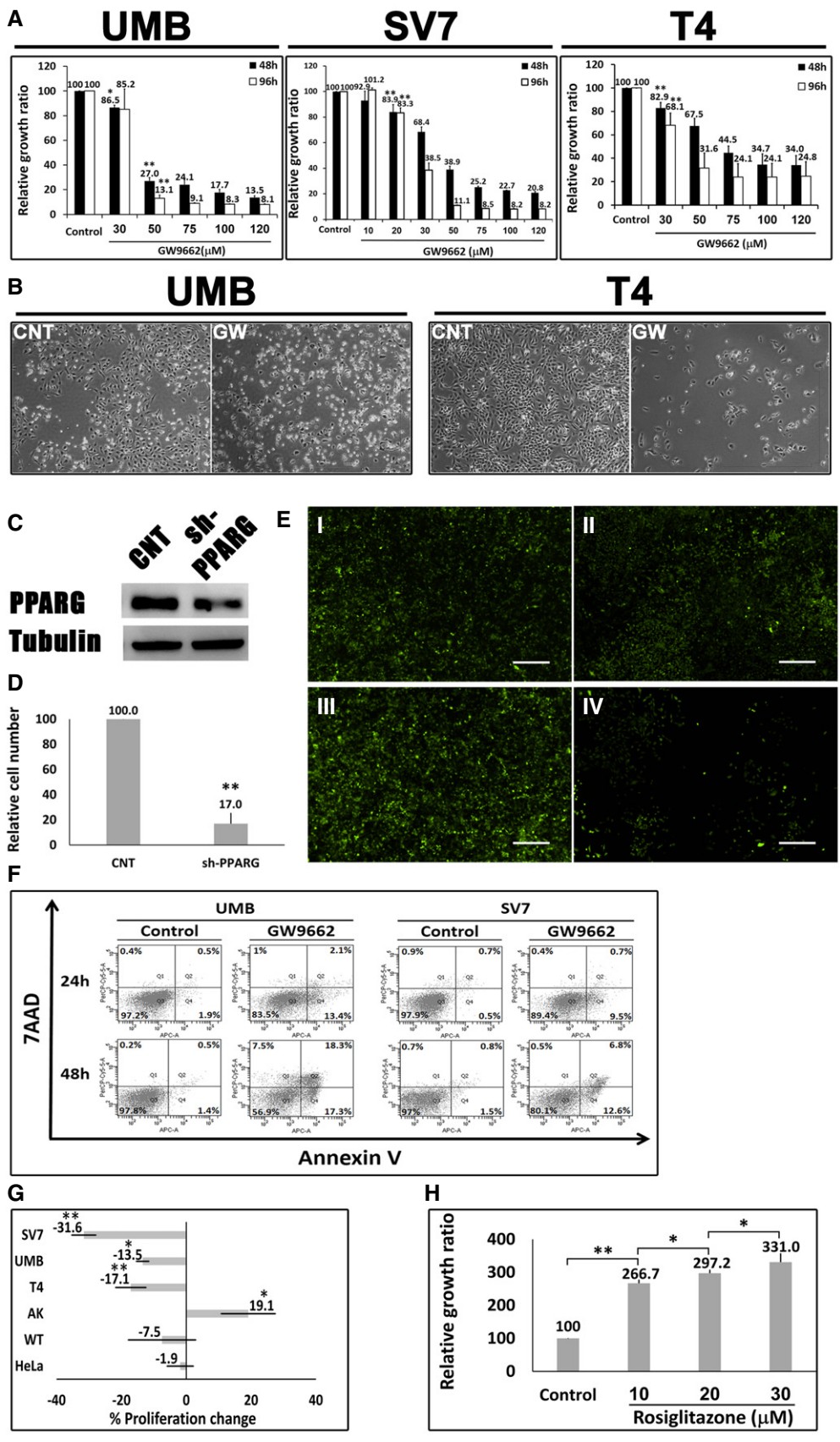

**Figure 4.**

**Figure 4. PPARG inhibition reduces the proliferation of AML cells.**

A   Relative growth ratios of UMB, SV7, and T4 (4th generation)-xenograft (Xn) cells (left, middle, and right panels, respectively) treated with the PPARG inhibitor GW9662 in varying concentrations, compared to control, vehicle-treated cells, shown as mean ± SD (n = 3). At each of the two time points, treated cells with a given concentration were compared to control cells. *P < 0.05, **P < 0.01 (one-way ANOVA with Bonferroni *post hoc* test).

B   The effects of 48-h treatment with GW9662 (GW) or vehicle (CNT) on the morphology of UMB (left panel) and T4-Xn (right panel) cells. Scale bar: 1,000 μm.

C   *PPARG* knockdown (KD) results in growth inhibition of AML cells: UMB and SV7 cells were infected with shRNA targeting *PPARG* or control-shRNA, both consisting of a GFP-expressing cassette. Western blot analysis demonstrated significant reduction in PPARG protein levels following infection with *PPARG*-shRNA (sh-PPARG) compared to control-shRNA (CNT).

D   In order to assess the effect of *PPARG* KD on cell growth, both UMB and SV7 cell lines were seeded in equal amounts and recounted following 1 week. *PPARG* KD resulted in a significant reduction in cell numbers. Results shown as mean ± SD (n = 3). **P < 0.01 (two-tailed Student's *t*-test).

E   Representative images of control- and *PPARG*-shRNA-infected cells, both marked by GFP, at day 0 and day 7, showing significant growth inhibition in the latter. I & II, control cells at day 0 and 7, respectively; III & IV, *PPARG*-shRNA-infected cells at day 0 and 7, respectively. Scale bars, 500 μm.

F   Flow cytometric analysis of UMB and SV7 cells treated with GW9662 or vehicle (control). Cells were treated for 24 or 48 h and analyzed for the expression of the early apoptosis marker annexin V and necrosis marker 7AAD.

G   Comparison of the anti-proliferative effect of GW9662 on UMB, SV7, T4-Xn, normal human adult kidney (AK), Wilms' tumor (WT), and the cervical cancer cell line HeLa, shown as percent change in proliferation. The cells were treated for 48 h with 30 μM GW9662. Results shown as mean ± SD (n = 3). *P < 0.05, **P < 0.01 (one-way ANOVA with Bonferroni *post hoc* test).

H   Relative growth ratios of T5-Xn cells treated with the PPARG agonist rosiglitazone for 96 h in varying concentrations, compared to control cells, shown as mean ± SD (n = 3). *P < 0.05, **P < 0.01 (one-way ANOVA with Bonferroni *post hoc* test).

Data information: The exact *P*-values are specified in Appendix Table S5.

any homozygous KO clone (0/14). These results imply that *PPARG* KO was technically successful, as indicated by the presence of *PPARG* heterozygous KO clones, and thus that complete *PPARG* KO may have resulted in strong growth inhibition leading to negative selection and accordingly inability to derive such clones. To determine whether PPARG inhibition reduces the proliferation of AML cells via the induction of apoptosis, we next stained UMB and SV7 cells treated with GW9662 for the early apoptosis marker annexin V and necrosis marker 7-AAD. Flow cytometry demonstrated that PPARG inhibition induces apoptosis in AML cells, which is already noticeable at 24 h and becomes more pronounced at 48 h, in accordance with the MTS assay (Fig 4F). In order to assess whether the effects of PPARG inhibition are specific to AML, we compared the effect of a 48-h treatment with GW9662 on the growth of AML cells (UMB, SV7, and T4-Xn) to its effect on the cervical cancer cell line HeLa and primary cells derived from the renal neoplasm Wilms' tumor (WT). Whereas all three types of AML cells demonstrated significantly lower proliferation, both HeLa and WT cells were unaffected (Fig 4G). Furthermore, when the same treatment regimen was applied to normal human kidney cells derived from different donors, no anti-proliferative effect was noted, even at high concentrations (Fig EV2). Rather, a slight increase in proliferation was noted (Fig 4G). To further assess the importance of PPARG signaling in AML propagation, we treated UMB, SV7, and T5-Xn cells with the PPARG agonist rosiglitazone. Following 48 h, UMB and SV7 cells demonstrated a minimal, statistically insignificant increase in cell growth compared to control, vehicle-treated cells, as assessed by the MTS assay, while T5 cells exhibited a significant increase in cell growth (Fig EV3). To better characterize this effect, we treated T5-Xn cells for 96 h with varying concentrations of rosiglitazone and measured cell viability compared to vehicle-treated cells via the MTS assay. Consequently, T5-Xn demonstrated a dose-dependent increase in proliferation (Fig 4H), further supporting the notion that AML-Xn are PPARG dependent. Taken together, these results demonstrate that human renal AML cells are PPARG dependent and highly sensitive to PPARG antagonism, which induces AML cell death, at least in part via induction of apoptosis. Importantly, this effect is specific to AML and is not seen when normal human kidney cells are exposed to the same treatment.

## PPARG inhibition damages AML cells at the structural and functional levels

Having demonstrated the anti-proliferative effect exerted by PPARG inhibition on AML cells, we subsequently wished to assess whether the same treatment would also affect AML cell function. First, we attempted to determine whether PPARG inhibition affects the migration capacity of AML cells. Toward this, UMB and T4-Xn cells were treated with GW9662 or vehicle for 48 h and then tested for their ability to migrate through a scrape made in the cell monolayer (Fig 5A). While control, vehicle-treated cells easily migrated through the scrape, GW9662-treated cells failed to do so within 48 h, indicating that PPARG inhibition reduces the migration capacity of AML cells. Since colony formation represents an important trait of both normal and tumor stem cells, we subsequently tested whether PPARG inhibition affects the ability of AML cells to form single-cell-derived colonies. A 48-h treatment with GW9662 significantly reduced the ability of both UMB and T4-Xn cells to form colonies when seeded at densities of 0.3 and 1 cell per well, compared to vehicle-treated cells (Fig 5B). Of note, PPARG inhibition completely abolished the colony formation capacity of Xn cells (Fig 5B), again suggesting that the proliferation and survival of Xn cells are highly PPARG dependent. In addition, PPARG inhibition markedly damaged the cytoskeletal structure of both UMB and Xn cells, as reflected by IF for β-actin (Fig 5C). Taken together, these results demonstrate that PPARG inhibition in AML cells results in significantly impaired migration and clonogenic potential, two key functional properties of tumor cells.

## PPARG inhibition impedes the *in vivo* tumor initiation capacity of AML cells

In light of the significant *in vitro* anti-proliferative effects of PPARG inhibition on AML cells, we sought to determine whether inhibition of the PPARG pathway could also limit the ability of AML cells to generate tumors *in vivo* (i.e., tumor initiation capacity). To this end, we used a previously described tumor engraftment assay (Bar *et al*, 2007; Fan *et al*, 2010), whereby cells are first pre-treated *ex vivo* and then injected into immunodeficient mice. In order to allow *in vivo*

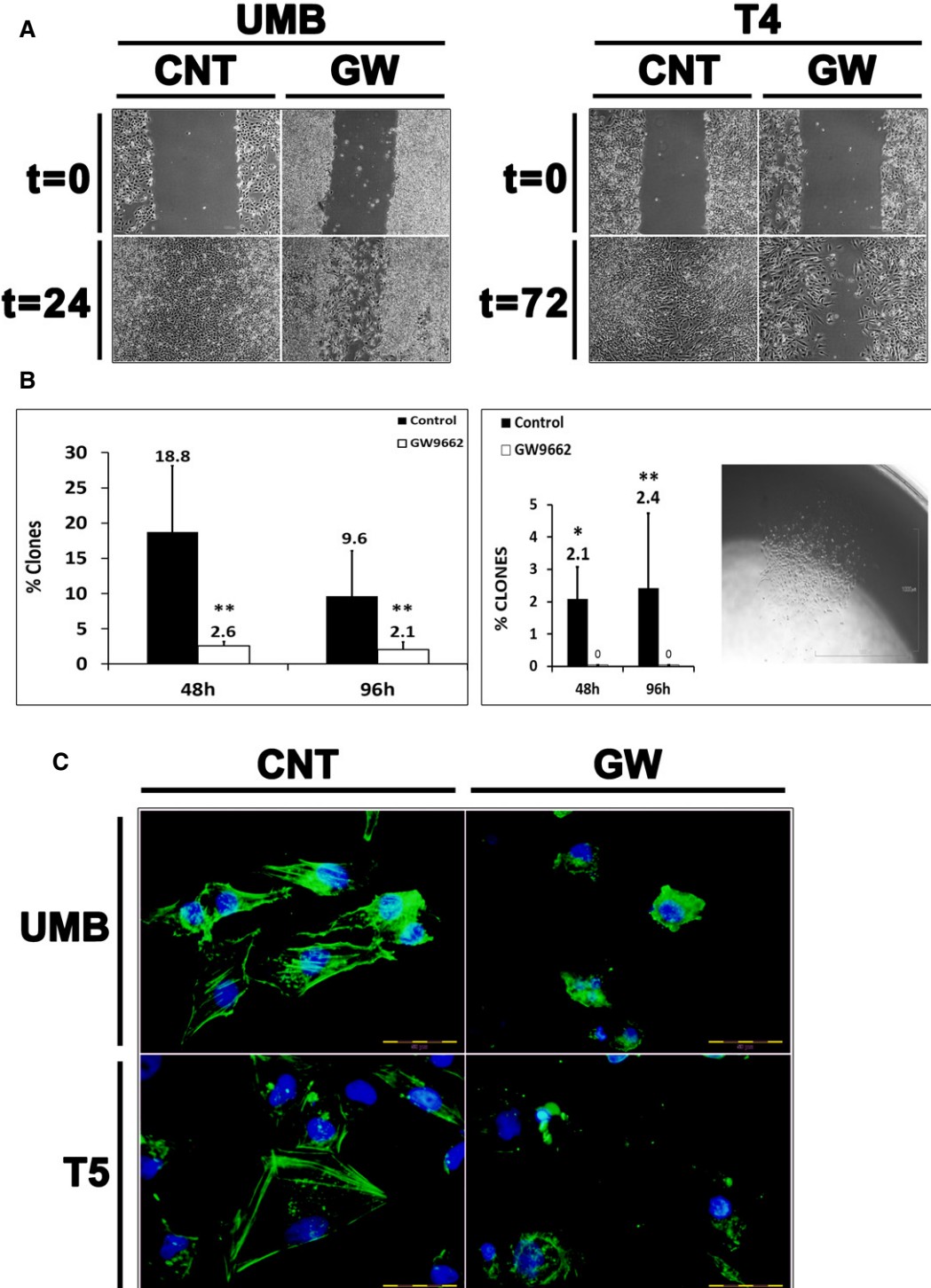

**Figure 5.  PPARG inhibition damages AML cells at the structural and functional levels.**

A  Cell migration assay demonstrating the effects of PPARG inhibition on cell migration. UMB and T4 (4th generation)-xenograft (Xn) cells were treated with GW9662 (GW) or vehicle (CNT) and assessed for their ability to migrate through a scrape in the cell monolayer. Both UMB (left panel) and T4-Xn cells (right panel) demonstrated significantly lower migration capacity as a result of the treatment, compared to control cells. Scale bar: 1,000 μm.

B  Comparison of the single-cell-derived colony formation capacity of UMB (left panel) and T4-Xn cells (right panel) treated with GW9662 or vehicle (control). Shown in the picture is a representative colony derived from a T4-Xn cell. Results shown as the mean percentage of cells that formed clones ± SD ($n = 3$). *$P < 0.05$; **$P < 0.01$ (chi-squared test). Scale bar: 1,000 μm. The exact $P$-values are specified in Appendix Table S5.

C  Immunofluorescent staining for β-actin (green) in UMB (top panel) and T5-Xn cells (bottom panel) treated with GW9662 (GW) or vehicle (CNT), demonstrating changes in the cytoskeleton as a result of the treatment. Scale bar: 50 μm.

       

imaging of tumor growth, we labeled AML-Xn cells with mCherry, using a retroviral vector. The cells were then treated for 24 h with GW9662 or vehicle, after which 50,000 living cells, as determined by negative staining for trypan blue, of each group, were injected into four mice. Starting from day 7 post-injection and every 4 days thereafter, the tumor burden was visualized using a fluorescent binocular and quantified as the percentage of visualized field

occupied by the tumor. The predetermined experimental end-point was set as the appearance of tumors requiring mouse scarification. On all three time points assessed (days 7, 11, and 15 post-injection), the tumor burden was significantly higher in the control group compared to the treatment group (Fig 6A and B). Specifically, at day 7, in all four control mice, a well-defined tumor was both visualized and palpable, while no tumor could be detected in the treatment

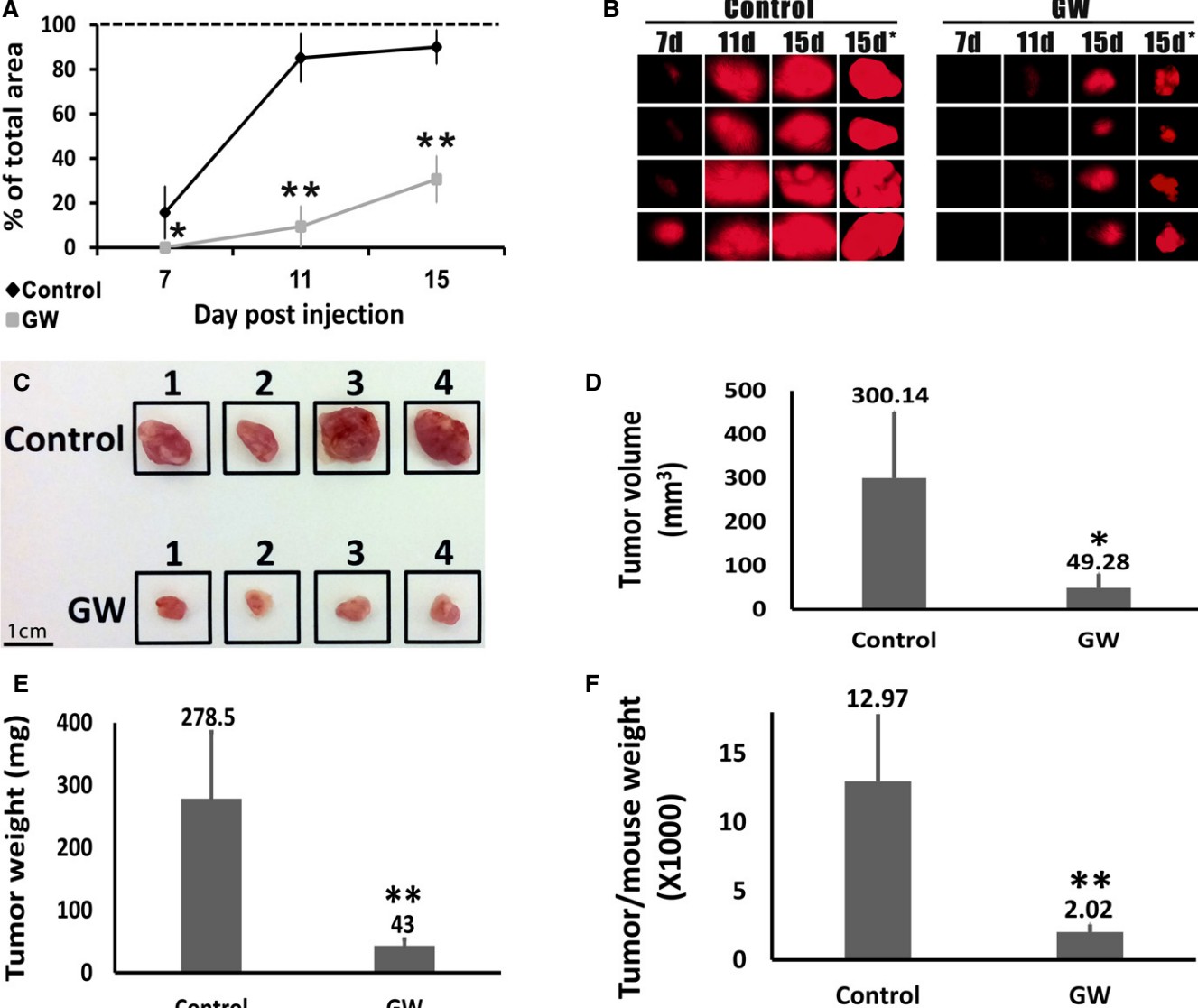

**Figure 6.  PPARG inhibition halts the *in vivo* tumor-initiation capacity of AML cells.**

mCherry-labeled AML-Xn cells were treated for 24 h with GW9662 or vehicle, after which 50,000 living cells, as determined by negative staining for trypan blue, of each group, were injected into four mice. Tumor burden was visualized at post-injection day 7 and every 4 days thereafter, using a fluorescent binocular and quantified as the percentage of visualized field occupied by the tumor.

A    Relative size of tumors derived from control- and GW9662-treated (GW) cells, as assessed by *in vivo* imaging. Size shown as percentage of the visualization field in the fluorescent binocular occupied by the tumor. Results shown as mean ± SD (*n* = 4). *$P < 0.05$; **$P < 0.01$ (two-tailed Student's *t*-test).

B    Photographs of the mCherry-expressing tumors obtained using a fluorescent binocular at 7, 11, and 15 days after injection and immediately after tumor resection (7, 11, 15, and 15 days*, respectively). Photographs taken at 0.7× magnification.

C    Photographs of the tumors derived from control and GW9662-treated (GW) cells, immediately following resection. Scale bar: 1 cm.

D–F  Comparison of the tumors derived from control and GW9662-treated (GW) cells in terms of tumor volume, tumor weight, and tumor weight to mouse weight. Results shown as mean ± SD (*n* = 4). *$P < 0.05$; **$P < 0.01$ (two-tailed Student's *t*-test).

Data information: The exact *P*-values are specified in Appendix Table S5.

mice (Fig 6A and B). The experiment was terminated at day 15 due to the large size of some of the tumors, requiring mouse euthanasia. In accordance with the imaging results, the tumors resected from control mice were significantly larger than those from the treatment mice (Fig 6B–D). Importantly, tumor size, as assessed by *in vivo* imaging was highly similar to that seen after resection (Fig 6B and C). Accordingly, the control tumors demonstrated a significantly higher weight, even when normalized to mouse weight (Fig 6E and F). Taken together, these results indicate that PPARG inhibition is highly effective in abrogating the tumor-initiation capacity of AML tumors.

## PPARG inhibition leads to AML cell death via TGFB1-mediated downregulation of PDGFB and CTGF

In order to decipher the mechanism mediating the effects of PPARG inhibition on AML cells, we carried out global gene expression analysis using RNA-sequencing, comparing UMB and SV7 cells treated for 12 or 24 h with GW9662 to vehicle-treated, control cells. Principal components analysis (PCA) of genes demonstrating a ≥ twofold

change revealed a similar effect of PPARG inhibition on both AML cell lines (Fig 7A). Importantly, this effect was similar when the 12 or 24 h treatment protocols were used. Interestingly, three of the most significantly downregulated genes were *PDGFB*, *CTGF*, and *PDGFRB*, all representing cardinal regulators of MSCs/pericytes (Fig 7A) (Hellstrom *et al*, 1999; Kale *et al*, 2005). IPA analysis demonstrated that treated AML cells exhibit several significantly inhibited cellular functions. These included reduced cell movement, vasculogenesis, proliferation of vascular smooth muscle cells, and cell survival (Table 2). Similarly, enriched GO functions in GW9662-treated AML cells included cell motion, regulation of angiogenesis, cell morphogenesis, and regulation of cell proliferation (Appendix Table S3). Detailed network analysis of several key functions decreased in AML cells following PPARG inhibition is shown in Appendix Figs S6–S9. In order to pinpoint the cellular pathways regulating these effects of PPARG inhibition on AML cells, we carried out IPA analysis of significantly changed upstream regulators. This analysis uncovered TGFB1 as the most significantly and strongly inhibited upstream regulator in PPARG-inhibited AML cells (Appendix Table S4). Accordingly, the second most inhibited

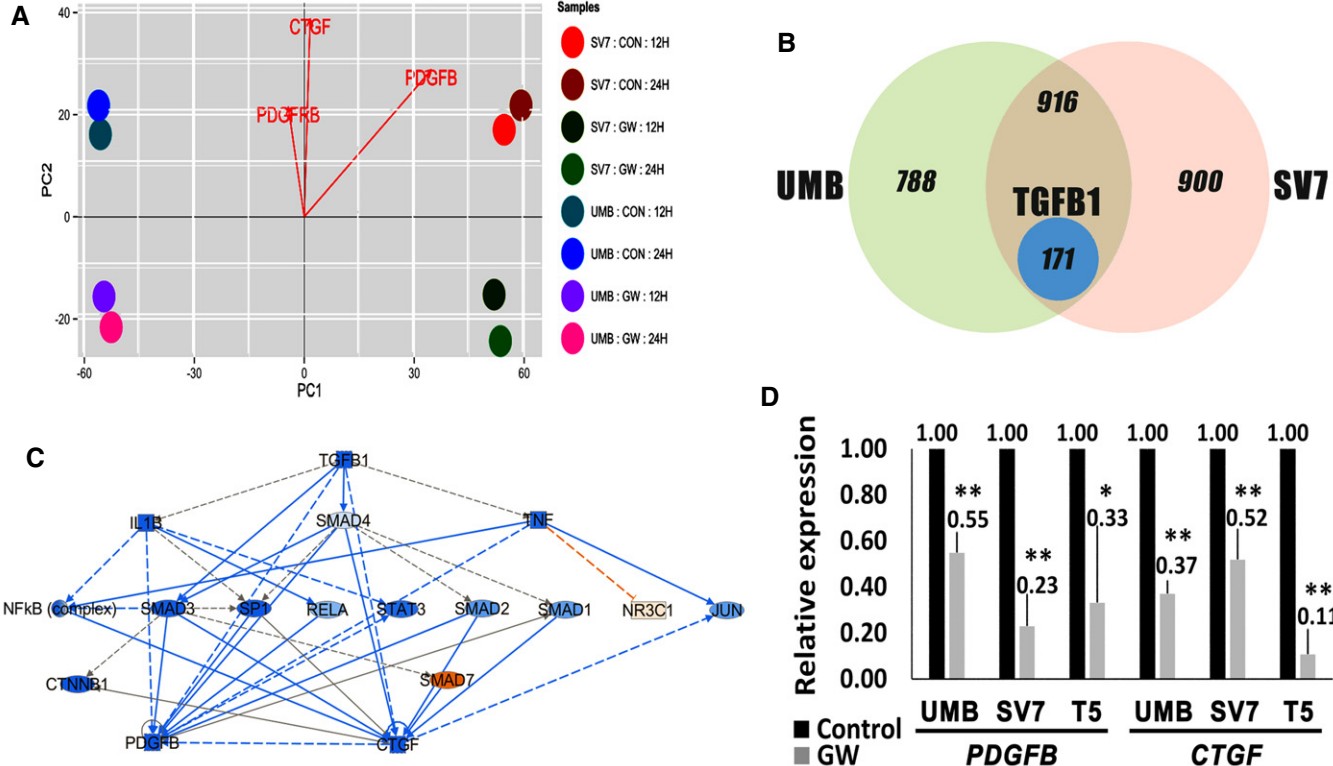

**Figure 7. PPARG inhibition induces AML cell death via TGFB1-mediated inhibition of PDGFB and CTGF.**

A  Principal components analysis (PCA) of differentially expressed genes in UMB and SV7 cells treated with GW9662 (GW) for 12 or 24 h compared to control cells (CON). Notably, *PDGFB*, *PDGFRB*, and *CTGF* demonstrate significantly stronger expression in control samples.

B  Venn diagram demonstrating the effect of PPARG inhibition on global gene expression of UMB and SV7 cells. The left and right circles represent the number of genes exhibiting a fold change of ≥ 2 as a result of PPARG inhibition in UMB and SV7 cells, respectively, while the intersection area represents the common differentially expressed genes. The blue circle represents the TGFB1-related genes contained within the latter.

C  IPA of TGFB1-regulated genes demonstrating a fold change of ≥ 2 as a result of PPARG inhibition. Blue and red circles represent inhibited and activated upstream regulators, respectively.

D  qPCR analysis of *PDGFB* and *CTGF* in UMB, SV7, and T5-Xn cells treated with GW9662 (GW) compared with control cells. Results shown as mean ± SD (n = 4). *P < 0.05; **P < 0.01 (two-tailed Student's t-test). The exact P-values are specified in Appendix Table S5.

**Table 2.    Ingenuity Pathway Analysis (IPA) demonstrating significantly inhibited cellular functions in angiomyolipoma (AML) cells treated for 24 h with GW9662 compared with control, vehicle-treated cells.**

| Function | P-value | # Molecules |
|---|---|---|
| Cell movement | 2.34E-21 | 268 |
| Vasculogenesis | 8.93E-17 | 122 |
| Migration of endothelial cells | 2.54E-11 | 55 |
| Proliferation of vascular smooth muscle cells | 1.11E-09 | 32 |
| Organization of cytoskeleton | 1.25E-09 | 148 |
| Movement of vascular smooth muscle cells | 2.10E-09 | 26 |
| Organization of cytoplasm | 2.98E-08 | 152 |
| Cell survival | 1.11E-07 | 154 |
| Cell transformation | 3.98E-06 | 60 |
| Remodeling of blood vessel | 6.31E-06 | 15 |

regulator was SMAD3, the main TGFB1 effector (Nakao *et al*, 1997). Other significant changes in the activity of upstream regulators included inhibition of PDGF BB and VEGF (Appendix Table S4), two important regulators of pericyte activity and vasculogenesis in both healthy and tumor tissues (Benjamin *et al*, 1998; Yang *et al*, 2013a, b; Cao, 2014; Ji *et al*, 2014; Iwamoto *et al*, 2015). Dissection of significantly changed genes revealed that 1875 and 1987 genes demonstrated a fold change of ≥ 2 in UMB and SV7 cells, respectively (Fig 7B). Of these, 1,087 genes were significantly changed in both cell lines, representing the effect of PPARG inhibition in AML cells regardless of the specific cell line. In accordance with the upstream regulator analysis, 171 of these 1,087 genes (approximately 15.7%) were found to be TGFB1 related (Fig 7B). IPA analysis of TGFB1-related genes in treated versus control cells, demonstrated significant inhibition in the activity of the TGFB1 effectors SMAD1, SMAD2, and SMAD3 alongside activation of SMAD7. Downstream to the TGFB1 pathway, we detected significant inhibition of PDGFB and CTGF (Fig 7C). In order to validate these findings, and assess whether they also apply to Xn cells, we carried out qPCR analysis of UMB, SV7, and T5-Xn cells treated with GW9662. In accordance with the RNA-sequencing results, we detected significant downregulation of both factors in both cell lines as well as in the Xn cells (Fig 7D). In conclusion, these results suggest that PPARG is upstream of the TGFB1 pathway and may mediate the anti-tumorigenic effects of PPARG inhibition.

**The AML cell of origin is likely a renal MSC-like cell**

Having established that UMB cells give rise to renal AML-like tumors *in vivo*, we wished to identify the renal cell type to which UMB cells correspond, as this would likely represent the AML cell of origin. AML is composed of several mesenchymal cell types and is histologically defined as a "PEComa" (i.e., arises from a perivascular cell type) (Folpe & Kwiatkowski, 2010). Hence, we hypothesized that it might originate from the renal population of perivascular cells, otherwise known as renal MSCs, which possess progenitor traits and

fit within standard criteria (Dominici *et al*, 2006). In order to test this hypothesis, we first used flow cytometry to examine AML cell lines for the expression of MSC surface markers (Fig 8A). Both UMB and SV7 cells demonstrated very high expression levels of the MSC markers CD73, CD90, and CD105, but did not express the hematopoietic markers CD3, CD45, CD34, and CD14. NCAM1 (CD56), previously described in MSCs (Buhring *et al*, 2009), showed moderate expression levels (23–55%). Subsequently, we tested the ability of AML cells to differentiate *in vitro* along mesodermal lineages via exposure to appropriate differentiation media. Culturing of both AML cell lines in adipogenic and osteogenic media for 3 weeks resulted in differentiation toward both lineages, as evident by the formation of perinuclear lipid vacuoles (Oil Red O staining) and by the positive Alizarin red staining, respectively (Fig 8B). We next wished to ascertain that AML cells correspond to MSCs in terms of gene expression. Since the kidney harbors several cell lineages (e.g., epithelial progenitors and stromal progenitors) (Pleniceanu *et al*, 2010; Dziedzic *et al*, 2014), we compared the expression of lineage-specific renal genes in AML cells to Wilms' tumor [WT, known to arise from epithelial progenitors (Pleniceanu *et al*, 2010; Pode-Shakked *et al*, 2013; Shukrun *et al*, 2014; Pode-Shakked *et al*, 2016‡)], human fetal kidney (hFK), and human MSCs via qPCR. WT expressed high levels of the entire renal epithelial progenitor gene set, including *OSR1*, *SIX2*, *CITED1*, and *PAX2*, compared to hFK (Fig 8C). In contrast, when compared to hFK, AML cells exclusively over-expressed *OSR1* (Fig 8C), with SV7 demonstrating comparable expression levels of *SIX2*. Of note, both OSR1 and SIX2 have been previously shown to be expressed in mesenchymal tissues (Fogelgren *et al*, 2008; Stricker *et al*, 2012). Hence, their expression in AML cells likely represents the mesenchymal nature of the cells rather than an epithelial progenitor identity. Importantly, *FOXD1*, the specifying gene of renal stromal progenitors (Pleniceanu *et al*, 2010), also showed low expression levels in AML cells compared to hFK, suggesting that AML does not originate from this cell type. Next, we evaluated the expression of lineage-specific renal genes in human MSCs compared to hFK. Analysis of gene expression in MSCs revealed a similar expression pattern to AML cells (Fig 8C), demonstrating relatively high expression levels of *OSR1* and *SIX2*, low expression levels of other renal epithelial progenitor genes (i.e., *PAX2* and *CITED1*), and comparable levels of *FOXD1*. Taken together, these results indicate that AML cells markedly differ in renal gene expression from both renal epithelial and stromal progenitors, but rather show a relatively similar expression pattern to MSCs. In order to explore the mechanisms involved in AML initiation, under the working hypothesis that MSC is the AML cell of origin and UMB cells represent their transformed counterpart, we carried out global gene expression analysis, comparing UMB cells to MSCs. IPA of enriched functions in UMB cells compared to MSCs demonstrated significant changes in several molecular functions (Fig 8D). These included increased proliferation and colony formation (consistent with transformation) and increased formation of vessels and adipocytic differentiation (consistent with the characteristic AML lineages) in UMB cells, compared to MSCs. Decreased functions included quantity of chondrocytes, ossification of bone, and attachment of cells (Fig 8D). These results support the

---

‡Correction added on 3 April 2017 after first online publication: references Pode-Shakked *et al*, 2013; Shukrun *et al*, 2014; Pode-Shakked *et al*, 2016 have been added.

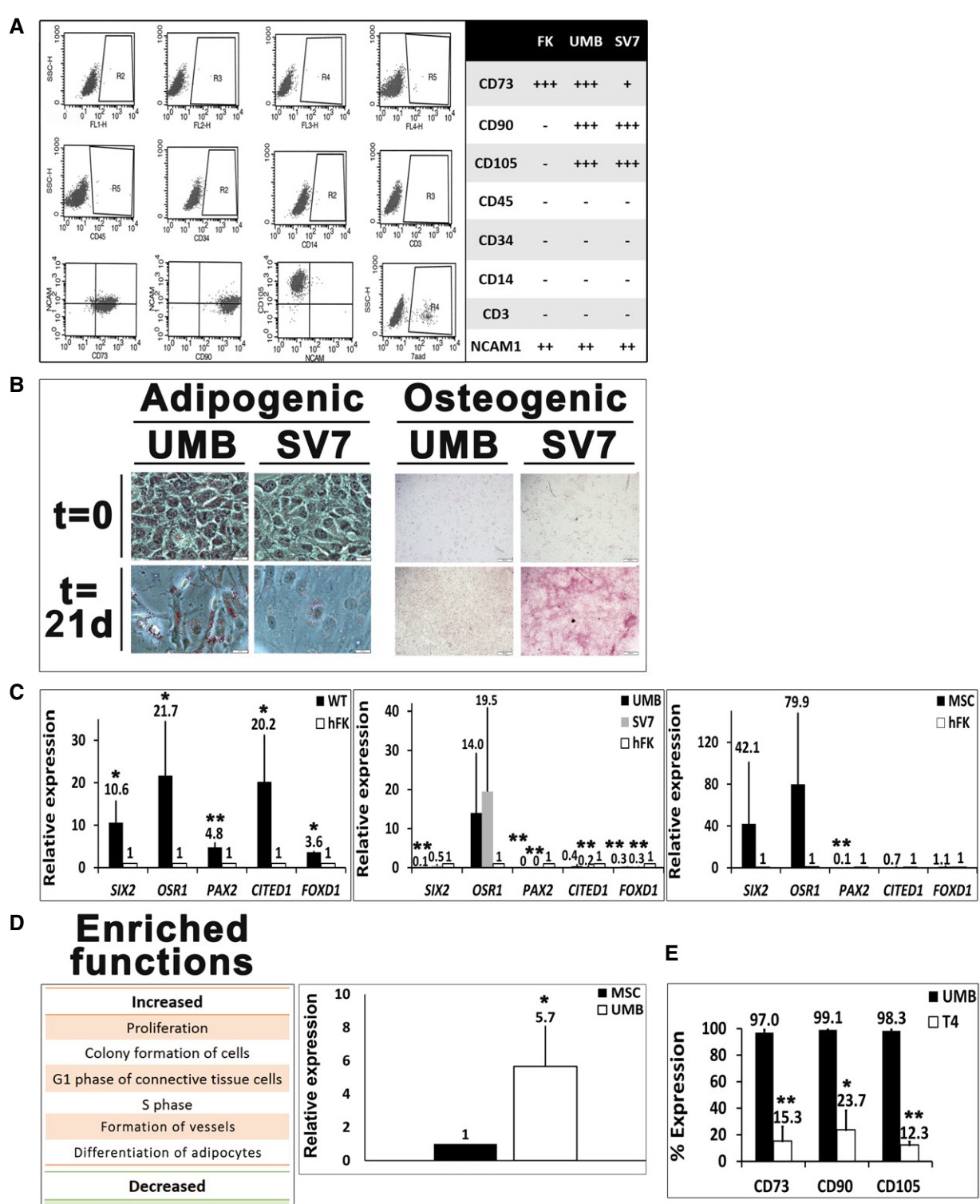

Figure 8.

**Figure 8.  The AML cell of origin is a renal mesenchymal stem cell (MSC)-like cell.**

A    Flow cytometry analysis of human fetal kidney (hFK), UMB, and SV7 cells. Left panel: Representative dot plots of UMB cell expression of CD45, CD34, CD14, CD3, CD105, CD73, CD90, neural cell adhesion molecule 1 (NCAM1), and unstained controls. Right panel: Summary of expression levels. +++, strong; ++, medium; +, low; −, no staining.

B    Left: Oil Red O staining of UMB and SV7 cells before ($t = 0$) and after ($t = 21$ days) induction of adipogenic differentiation, demonstrating formation of lipid vacuoles. Scale bar: 50 μm. Right: Alizarin red staining of UMB and SV7 cells before ($t = 0$) and after ($t = 21$ days) induction of osteogenic differentiation, demonstrating formation of calcium deposits. Scale bar: 500 μm.

C    Left panel: Quantitative PCR (qPCR) analysis, demonstrating significant upregulation of renal progenitor genes in Wilms' tumor (WT) cells compared to hFK cells (hence = 1). Middle panel: qPCR analysis of renal progenitor gene expression, demonstrating higher *OSR1* expression in UMB and SV7 cells and comparable *SIX2* expression in SV7 compared to hFK (hence = 1). Other renal progenitor genes and the stromal progenitor gene *FOXD1* exhibit low expression levels. Right panel: qPCR analysis of renal progenitor gene expression in human MSCs compared to hFK (hence = 1), demonstrating a similar expression pattern to AML cells. Results shown as mean ± SD ($n = 3$). *$P < 0.05$; **$P < 0.01$ (two-tailed Student's *t*-test).

D    Left panel: Ingenuity Pathway Analysis (IPA) of increased and decreased functions in UMB cells compared to MSCs. Right panel: qPCR analysis of *PPARG* expression in UMB cells compared with MSCs (hence = 1). Results shown as mean ± SD ($n = 3$). *$P < 0.05$ (two-tailed Student's *t*-test).

E    Comparison of MSC marker expression between UMB and T4 (4$^{th}$ generation)-xenograft (Xn) cells. A significant decrease in the expression of CD73, CD90, and CD105 is seen. Results shown as mean ± SD ($n = 3$).*$P < 0.05$; **$P < 0.01$ (two-tailed Student's *t*-test).

Data information: The exact *P*-values are specified in Appendix Table S5.

notion that AML initiation, as reflected by the presumed transition from the normal renal MSC into an AML-like cell, involves activity of molecular circuits supporting tumorigenic proliferation of MSCs and their differentiation into the specific lineages seen in AML, at the expense of other MSC lineages (i.e., bone and cartilage). Interestingly, qPCR analysis revealed higher *PPARG* expression in UMB cells compared to human MSCs (Fig 8D), implying that PPARG activation may also be involved in AML initiation. Finally, we wished to determine whether the cells that propagate AML-Xn *in vivo* retain MSC marker expression. We therefore analyzed the expression of CD73, CD90, and CD105 via flow cytometry. We found that $15.30 \pm 10.79\%$, $23.73 \pm 14.5\%$, and $12.3 \pm 2.46\%$ of cells stained for CD73, CD90, and CD105, respectively (Fig 8E). This implies that the cells that propagate human AML are phenotypically different from the tumor-initiating cells, being diverted from the "classical" MSC phenotype. Taken together, these results indicate that human AML cells, capable of initiating AML-like tumors *in vivo*, are in fact renal MSC-like cells.

# Discussion

The peculiar histology of AML has been the subject of much investigation. While previously hypothesized to represent a hamartoma, later studies disclosed the clonal nature of AML (Kattar *et al*, 1999). It is currently accepted that renal AML results from aberrant differentiation of a transformed renal progenitor (Folpe & Kwiatkowski, 2010). Accordingly, two key questions surround AML biology. First, which molecular mechanisms drive its development and growth? Second, what is the AML cell of origin? Unfortunately, the study of AML, the leading cause of death in TSC, is severely limited by the absence of an *in vivo* model, thereby hampering the development of new therapies. Most studies attempting to develop such a model relied on TSC1/2-deficient rodents (Kobayashi *et al*, 1995, 1999, 2001; Liang *et al*, 2014). However, none has resulted in tumors exhibiting the full histological features of genuine AML. Via serial propagation of human AML cells in mice, we provide the first description of such a model. Despite the use of immunodeficient mice, which eliminates potential relevance of immunologic effects on tumor development, this model harbors two inherent advantages. First, it fully recapitulates AML, as evident by the triphasic

histology, myo-melanocytic phenotype, and mTORC1 activation. Second, the tumors represent human AML tissue and are thus an invaluable tool to study human AML pathogenesis, uncover new molecular pathways involved in its growth, and test new drugs. Analysis of gene expression changes along Xn propagation *in vivo* and in comparison with normal human kidney, uncovered PPARG as a regulator of tumor growth. Consequently, we show that PPARG inhibition may serve as an effective therapy for both sporadic and TSC-related AMLs. While PPARG has been classically regarded as a master regulator of adipogenesis (Lehrke & Lazar, 2005), more recent studies have implicated it in various cellular processes, including proliferation, apoptosis, angiogenesis, and cancer (DuBois *et al*, 1998; Wada *et al*, 2006; Tian *et al*, 2009; Meng *et al*, 2011; Yuan *et al*, 2012). Interestingly, initial studies used PPARG *agonism* as anti-neoplastic therapy (Tontonoz *et al*, 1997; Mueller *et al*, 1998). However, despite some positive results in several tumor types, PPARG agonists are not widely used due to lack of efficacy. More recently, however, PPARG antagonism was successfully used to halt the *in vitro* growth of tumor cells of various sources [e.g., breast (Yuan *et al*, 2012; Wang *et al*, 2013), esophagus (Takahashi *et al*, 2006), and pancreas (Nakajima *et al*, 2008)]. Although it is unclear why PPARG was selected as a therapeutic target in these studies, it has been previously proposed that PPARG activation may promote tumor survival by converting toxic fatty acids into inert triglycerides (Kourtidis *et al*, 2009). In contrast, we *prospectively* identified PPARG activation at the center of a molecular network accompanying AML-Xn growth *in vivo*. Similar results were obtained in a cohort of primary AML tumors, which uniformly exhibit robust PPARG expression. Of note, PPARG activation in our model is not merely a surrogate of adipogenic differentiation, which is usually seen in AML. First, adipocytes represent a relatively small proportion of tumor mass in late Xn generations (Appendix Fig S10). Second, detailed bioinformatic analysis showed that various genes regulating tumor growth, *non-adipogenic* differentiation, and the mTOR pathway are regulated by PPARG. Third, the robust and specific anti-proliferative effect of PPARG inhibition on AML cells implicates PPARG as a regulator of AML proliferation. Had PPARG activation during Xn propagation represent merely adipogenic differentiation, PPARG inhibitors would have been expected to block differentiation, thereby reciprocally increasing cell proliferation. Fourth, immunostaining revealed strong nuclear PPARG expression

in both the adipogenic and non-adipogenic components of AML-Xn, including the mass of undifferentiated epithelioid cells which likely drive tumor growth. Finally, strong PPARG expression was detected in both fat-containing and fat-poor AML. Thus, we propose that PPARG activation in AML does not reflect the characteristic adipogenic differentiation of this tumor. Instead, our results indicate that the presence of fat is secondary to PPARG being a master regulator of AML growth. The second unresolved issue, aside from the mechanisms driving AML growth, is the identity of its cell of origin. Despite being termed a "PEComa", indicating a perivascular cell of origin, the normal "PEC" counterpart has not been elucidated. Moreover, the unique AML mesenchymal differentiation pattern and distinctive "myo-melanocytic" phenotype, as well as the fact that it can arise in various organs, have added even more confusion as to the identity of its cell of origin. Based on these facts, we hypothesized that the AML cell of origin is the MSC/pericyte, which occupies a perivascular niche in virtually every tissue and harbors broad mesenchymal differentiation potential (Crisan et al, 2008). The ability to derive triphasic AML-Xn from UMB cells led us to characterize the latter, under the assumption that they represent a transformed equivalent of the AML cell of origin. Indeed, we found great similarity in surface marker expression, differentiation potential, and renal gene expression between UMB and SV7 cells and human MSCs. Notably, we found that blood vessels within AML-Xn consist primarily of mouse-derived endothelial cells. In contrast, adipogenic and myogenic cells, as well as undifferentiated epithelioid cells, were uniformly HLA$^+$. Collectively, a model of AML genesis can be envisioned, in which transformed pericytes take on an epithelioid phenotype and drive tumor growth, and at the same time give rise to the three main AML components. The vascular component seems to form when affected pericytes recruit normal endothelial cells, in a process that does not involve endothelial differentiation of tumor cells. In contrast, the adipogenic and myogenic compartments appear to result from direct differentiation of transformed pericytes. While it is unclear why different tumors harbor different proportions of each component, it is likely that the differentiation level of the epithelioid compartment dictates tumor aggressiveness, as tumors composed solely of epithelioid cells (i.e., epithelioid AML variant) are more aggressive (Cibas et al, 2001; Konosu-Fukaya et al, 2014). Moreover, our results indicate that pericytes/MSCs, previously suggested to play a role in AML pathogenesis (Yan et al, 2012; Siroky et al, 2014) drive tumor growth, and are inherently skewed toward the specific lineages seen in AML (e.g., fat and smooth muscle) at the expense of other MSC lineages (e.g., bone and cartilage). This tendency is mediated, at least in part, by PPARG. Specifically, PPARG activation alongside WNT5A downregulation, seen during in vivo propagation, may represent a major determinant of AML pathogenesis, as these two factors are key regulators of MSC/pericyte fate decisions, and their unique expression status in AML could underlie the skewed MSC differentiation and consequently the cellular phenotypes characterizing AML. In support of MSC being the AML cell of origin, it has been previously shown that mTOR activation promotes adipogenic differentiation of human MSCs (Yu et al, 2008), while inhibition of this pathway promotes osteogenesis (Martin et al, 2010). mTOR activation was also shown to induce

myogenic differentiation (Shu et al, 2002). Notably, a subpopulation of MSCs during development arises from the neural crest (Takashima et al, 2007), which may account for the expression of melanocytic markers in AML. Conclusive proof of this model would require an in vivo pericyte lineage-tracing system. Currently, however, neither a specific marker for this heterogeneous population nor a transgenic mouse model of AML is available. Of relevance, recent reports have identified MSC as the cell of origin in several malignant mesenchymal tumors (Boeuf et al, 2008; Mohseny et al, 2009; Danielson et al, 2010; Riggi et al, 2010). Moreover, it is becoming increasingly clear that the tumor-initiating cell of many, if not most, tumors is a resident stem/progenitor cell (Alison et al, 2010). Considering the stem cell populations present in the kidney throughout fetal and postnatal periods (Harari-Steinberg et al, 2011; Harari-Steinberg et al, 2013[#]), MSCs seem the most reasonable in light of the mesenchymal nature of AML. Of note, first-generation Xn resemble malignant epithelioid AML (Konosu-Fukaya et al, 2014), being composed of sheets of HMB-45$^+$ highly proliferative epithelioid cells. The emergence of this well-described, albeit rare variant of, AML, might be the result of its derivation from a purified population of tumor-initiating cells (UMB cells). Accordingly, later-generation Xn show the classical triphasic appearance. This may result from the fact that the Xn originated from a specific cell type (i.e., UMB cells), which requires several in vivo passages to allow differentiation-related processes to take place and give rise to the histology seen at later passages. Surprisingly, this increased differentiation was accompanied by a paradoxical decrease in time intervals necessary for tumor growth. Hence, signaling events promoting differentiation can also be associated with increased tumor growth. We detected inhibition of TGFB1 and its downstream targets, PDGFB and CTGF, as the mechanism underlying the effect of PPARG inhibition on AML. TGFB1 signaling has been shown to promote proliferation, migration, and in vivo function of human MSCs (Jian et al, 2006; Luo et al, 2012; Gao et al, 2014). Intriguingly, TGFB1 activation was shown to enhance the in vivo pericyte phenotype acquired by MSCs (Mendel et al, 2013). Specifically, TGFB1 inhibition is associated with apoptosis of mesangial cells, representing resident renal pericytes. Concomitantly, CTGF is as a key regulator of angiogenesis, promoting TGFB-dependent proliferation, migration, and activation of pericytes, recruitment of endothelial cells, and vessel formation (Kale et al, 2005; Pi et al, 2011), while CTGF inhibition significantly reduces human MSC proliferation (Battula et al, 2013). Similarly, PDGFB, a master regulator of vasculogenesis during development and tumorigenesis (Cao et al, 2011; Xue et al, 2011; Cao, 2013), is known to promote human MSC proliferation and migration (Cheng et al, 2013; Sun et al, 2013) and regulate MSC recruitment and differentiation into a pericyte phenotype in tumors, thereby enhancing tumor angiogenesis (Abramsson et al, 2003; Furuhashi et al, 2004). PDGFB also regulates the migration and proliferation of pericytes and pericyte progenitors (Lindahl et al, 1997; Hellstrom et al, 1999), at least in part via the TGFB pathway (Nadal et al, 2002), acting synergistically with CTGF (Hall-Glenn et al, 2012). Notably, CTGF was shown to regulate renal angiogenesis by recruiting and activating perivascular renal mesenchymal (i.e., mesangial) cells (Hellstrom et al, 1999). Of

---

[#]Correction added on 3 April 2017 after first online publication: reference Harari-Steinberg et al, 2013 has been added.

    

interest, increased PDGFB expression has been linked to pulmonary capillary hemangiomatosis, characterized by excessive capillary proliferation (Assaad *et al*, 2007). Hence, it appears that PPARG inhibition, via TGFB1 downregulation, hampers the proliferation, as well as other cellular functions, of AML cells, which represent MSC-like cells. From the translational perspective, PPARG inhibition might serve as a new treatment for AML, given that mTORC1 inhibitors are unable to completely eradicate AML (Bissler *et al*, 2008, 2013) and were shown to result in significant side effects (Trelinska *et al*, 2015). Importantly, normal kidney cells were completely unaffected by PPARG inhibition, implying specific targeting of tumor cells. A theoretical concern regarding PPARG inhibition is a potential deleterious effect on glucose metabolism, as PPARG agonists are anti-diabetic drugs. However, such effect has never been demonstrated. In fact, GW9662 has an anti-obesity effect in mice (Nakano *et al*, 2006), while mice heterozygous for *PPARG* deficiency show improved insulin sensitivity (Miles *et al*, 2000). Similarly, humans harboring a substitution in the PPARG isoform, leading to lower PPARG activity, exhibit lower weight and better insulin sensitivity (Deeb *et al*, 1998).

In conclusion, we have successfully established an *in vivo* AML model that might serve to gain new insights into tumor biology and test potential treatments. Better understanding of the cellular origins and genetic identity of AML would undoubtedly have important implications on defining biological mechanisms involved in tumor progression, treatment decisions, and drug discovery, as previously shown for other tumors (Brannon *et al*, 2010). We propose a model of AML generation from pericytes and provide compelling preclinical evidence of the value of PPARG inhibition in the treatment of AML. In light of the growing body of evidence supporting an association between MSCs and sarcomas (Boeuf *et al*, 2008; Mohseny *et al*, 2009; Danielson *et al*, 2010; Riggi *et al*, 2010), our findings may well have broader clinical implications.

# Materials and Methods

### Cell lines

Two cell lines derived from two renal AML patients were used: "UMB", derived from a TSC-related tumor, and "SV7", derived from a sporadic tumor (Lim *et al*, 2007).

### *In vivo* xenografting experiments

$10^6$ UMB and SV7 cells were subcutaneously injected in 100 μl of 1:1 medium:Matrigel (BD Biosciences, San Jose, CA, USA) into the flanks of 5- to 8-week-old NOD/SCID mice. UMB cells formed Xn after approximately 4.5 months. Xn were removed when they reached a diameter of 1.5 cm. Single-cell suspensions were obtained by mincing the samples in Iscove's modification of Dulbecco's medium (IMDM) containing antibiotics (penicillin and streptomycin), followed by treatment with collagenase-IV for 2 h at 37°C. Enzymatically treated tissue was triturated using IMDM at twice the volume of the collagenase solution and the suspension was filtered (100-μm cell strainer) and washed twice with IMDM containing antibiotics. Serial transplantation of dissociated cells from freshly

retrieved Xn was performed by injecting $10^6$ cells in 100 μl 1:1 serum-free medium:Matrigel subcutaneously into the flanks of 5–8-week old NOD/SCID mice.

### Microarray experiments

All chip array experiments were performed using Affymetrix Prime-View. Total RNA was extracted using TRIzol reagent (Life Technologies) according to the manufacturer's instructions. RNA samples were used to prepare biotinylated target DNA, according to the manufacturer's instructions. The target complementary DNA (cDNA) generated from each sample was processed as per manufacturer's recommendation using an Affymetrix GeneChip Instrument System (www.affymetrix.com/support/downloads/manuals/wt_sensetarget_label_manual.pdf). RNA quality and amount were confirmed using an agarose gel or by Bioanalyzer (Agilent). After scanning, array images were assessed by eye to confirm scanner alignment and the absence of significant bubbles or scratches on the chip surface. The signals derived from the array were assessed using various quality assessment metrics. Gene-level Robust Multi-array Average (RMA) sketch algorithm (Affymetrix Expression Console and Partek Genomics Suite 6.2) was used for crude data generation. Significantly changed genes were filtered as changed by at least 1.5-fold (*P*-value: 0.05). Predictions of functionality were based on gene ontology (GO) and IPA. The data discussed in this publication have been deposited in NCBI's Gene Expression Omnibus and are accessible through GEO Series accession number GSE94114 (https://www.ncbi.nlm.nih.gov/geo/query/acc.cgi?acc=GSE94114). For the comparison of AML-Xn to normal human kidney, data derived from GSE54227 were used for the latter.

### Assessment of cell viability

Cell viability was measured as previously described (Pode-Shakked *et al*, 2009).

### Apoptosis assay

Apoptosis assays were carried out as previously described (Pode-Shakked *et al*, 2009).

### Cell migration assay

Cells were grown overnight, after which the medium was changed and supplemented with 50 μM of GW9662 or DMSO, as control. After 48 h, a scrape was made through the confluent monolayers using a plastic pipette tip. Treated and control cells were then photographed at identical time points using Nikon Eclipse TS100.

### Clonogenicity assay

The cells' clonogenic capacity was assessed as previously described (Tropepe *et al*, 1999).

### Differentiation assays

Differentiation potential into mesenchymal lineages was assessed as previously described (Varda-Bloom *et al*, 2014).

## Statistics

Error bars represent the mean ± SEM, unless otherwise indicated. Statistical differences in gene expression (qPCR) and surface marker expression were evaluated using a non-paired two-tailed *t*-test. Statistical differences in clonogenic potential were determined using the chi-squared test. Cell viability assays and time necessary for xenograft growth were evaluated using one-way analysis of variance (ANOVA) followed by Bonferroni *post hoc* test. For all statistical analyses, the level of significance was set as $P < 0.05$. The exact *P*-values obtained in the experiments are shown in Appendix Table S5. Sample size was chosen based on previous results of mean and standard deviation of Xn size obtained in other tumor Xn experiments in similar conditions, requiring an alpha of 5%. All experiments were carried out in at least three biological replicates. The animals were randomized before cell injection for the treatment or control group. Data analysis was not blinded. Premature death was defined as a predetermined exclusion criteria.

## Tumor initiation assay

In order to assess the effect of PPARG inhibition on tumor initiation *in vivo*, we used a previously described tumor engraftment assay (Bar *et al*, 2007; Fan *et al*, 2010), whereby cells are first pre-treated *ex vivo* and then injected into immunodeficient mice. In order to allow *in vivo* visualization, AML-Xn cells were labeled with the fluorescent marker mCherry as previously described (Buzhor *et al*, 2013). Labeled cells were then seeded and treated with GW9662 or control vehicle. Following 24 h, $5 \times 10^4$ viable cells (as determined by negative staining with trypan blue) of each group were subcutaneously injected into NOD/SCID mice in 100 μl of 1:1 medium:Matrigel (BD Biosciences). For tumor burden assessment, mice were anesthetized and shaved, and subsequently visualized under Olympus SZX16 fluorescent binocular. Following sacrificing, tumors were removed and photographed and their size was assessed using a caliper and their weight was assessed using a Sartorius CP423S balance. Blinding was not applied.

## RNA-sequencing

Bulk total RNA was prepared from ~$1.5 \times 10^5$ cells using the Zymo Research's RNA Isolation kit (the Direct-zol™-96 RNA) according to the manufacturer's instructions. RNA was quantified on an Agilent BioAnalyzer, and 1 μg of RNA was made into cDNA libraries using the TruSeq mRNA-Seq library kit (Illumina). Libraries were sequenced on Illumina HiSeq 2000 to obtain $1 \times 50$ bases reads. Data from the high-throughput sequencing was analyzed based on the protocol by Anders *et al* (2013). Briefly, the reads were aligned by TopHat2 (Kim *et al*, 2013) to the human hg19 genome (the genome and the appropriate GTF files were obtained from the Illumina igenome collection), and the features were counted by HTSeq (Anders *et al*, 2015). Data normalization and differential gene expression were done by DESeq2 (Love *et al*, 2014). The data discussed in this publication have been deposited in NCBI's Gene Expression and are accessible through GEO Series accession number GSE94461 (https://www.ncbi.nlm.nih.gov/geo/query/acc.cgi?acc=GSE94461).

### The paper explained

#### Problem

Angiomyolipoma is a unique tumor that, despite a generally benign histology, can result in severe morbidity and mortality. Although mTORC1 activation has been implicated in its pathogenesis, AML only partially responds to mTORC1 inhibitors, emphasizing the need to uncover other causative factors. Similarly, while AML has been described as arising from a PEC, the physiological counterpart of the latter is enigmatic. Hence, two key questions surround AML: (i) Which molecular mechanisms govern its formation and growth? (ii) What is its cell of origin? Unfortunately, to date there is no *in vivo* model of human AML, which severely impedes the research of this tumor.

#### Results

In this paper, we establish an *in vivo* model of human AML in mice, mirroring the human tumor in histology, immunophenotype, and gene expression. We use this model to uncover PPARG as a key factor in the initiation and propagation of human AML-Xn *in vivo* and demonstrate that PPARG antagonism is an effective and specific strategy for targeting AML cells *in vitro* and preventing tumor initiation *in vivo*. We identify downregulation of the TGFB1 pathway, and consequently inhibition of PDGFB and CTGF, as the mediator of this effect. Finally, we provide evidence that the AML cell of origin is likely a resident renal MSC/pericyte.

#### Impact

The establishment of an *in vivo* model of human AML is important, as it provides a useful and robust tool both to identify novel molecular pathways underlying AML growth and test new therapies. Moreover, the uncovering of PPARG inhibition as an effective anti-AML treatment could serve as the basis for the development of more effective therapeutic strategies aimed at fully eradicating, rather than shrinking, AML tumors.

## Study approval

This study was conducted according to the principles expressed in the Declaration of Helsinki. The study was approved by the Institutional Review Board of Sheba Medical Center (SMC-9367-12) and Assaf Harofeh Medical Center (71/31) hospitals. All patients provided written informed consent for the collection of samples and subsequent analysis. All animal experiments were conducted in accordance with the National Institutes of Health guidelines for the care and use of animals and with an approved animal protocol from the Sheba Medical Center Animal Care and Use Committee. For all animal experiments, 5–8-week-old NOD/SCID mice, male and female, purchased from ENVIGO (previously Harlan laboratories), were used. All experiments were carried out in the pathogen-free animal facility of the Sheba Medical Center.

See Appendix Supplementary Methods for more information on cell culture, human FK and AK tissues, hematoxylin & eosin staining, IF staining of cells, flow cytometry, quantitative real-time RT-PCR analysis of gene expression, and IHC and immunofluorescent (IF) staining of paraffin-embedded Xn tissues.

**Expanded View** for this article is available online.

## Acknowledgements

BD is supported by the Ziering Foundation, the Israel Cancer Association (grant no 20150916), and the Israel Cancer Research Fund (ICRF) project

grant (grant number PG-14-112). JLA is supported by the grant RO1 AR47901 and P30 AR42687 Emory Skin Disease Research Core Center Grant from the National Institutes of Health, the Robert Margolis Foundation, as well as funds from the Rabinowitch-Davis Foundation for Melanoma Research and the Betty Minsk Foundation for Melanoma Research, and the Reynolds Sarcoma Foundation. We thank Dr. Jasmin Jacob of the Cancer Research Center, Sheba Medical Center, Tel Hashomer, Israel, for her assistance with bioinformatic analysis. We also wish to express our gratitude to Prof. Iris Barshack of the Department of Pathology, Sheba Medical Center, Tel Hashomer, Israel, for providing AML histology specimens, and assistance with stainings. We dedicate this work to our beloved friend and colleague, Klaudyna Dziedzic.

## Author contributions

OP, BD, and OH-S designed the experiments. OP, RShu, DO, MM-D, SP-C, and DDB-L performed the *in vivo* experiments. OP, RShu, DO, EV, KD, NP-S, YG, HA, NV-B, NB, RSht, LA, and AU performed the *in vitro* assays. OP, IK, and TK carried out the bioinformatic analyses. OP, BD, NV-B, and OH-S analyzed the data. AN and JLA provided the cell samples. OP, JLA, and BD wrote the manuscript.

## Conflict of interest

The authors declare that they have no conflict of interest.

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
