## [Review Process File · EMBO Molecular Medicine]

PPARG is central to the initiation and propagation of human angiomyolipoma, suggesting its potential as a therapeutic target

Oren Pleniceanu, Racheli Shukrun, Dorit Omer, Einav Vax, Itamar Kanter, Klaudyna Dziedzic, Naomi Pode-Shakked, Michal Mark-Daniei, Sara Pri-Chen, Yehudit Gnatek, Hadas Alfandary, Nira Varda-Bloom, Dekel D. Bar-Lev, Naomi Bollag, Rachel Shtainfeld, Leah Armon, Achia Urbach, Tomer Kalisky, Arnon Nagler, Orit Harari-Steinberg, Jack L. Arbiser and Benjamin Dekel

Corresponding author: Benjamin Dekel, The Pediatric Stem Cell Research Institute, Edmond & Lily Safra Children's Hospital, Sheba Center for Regenerative Medicine, Sheba Medical Center, Israel.

Review timeline:

Submission date:	29 November 2015
Editorial Decision:	07 January 2016
Revision received:	12 December 2016
Editorial Decision:	10 January 2017
Revision received:	19 January 2017
Editorial Decision:	26 January 2017
Accepted:	26 January 2017

Editor: Roberto Buccione

Transaction Report:

1st Editorial Decision

07 January 2016

Thank you for the submission of your manuscript to EMBO Molecular Medicine and please accept our apologies for the delay, due also to the concomitant holiday season.

We have now heard back from the three Reviewers whom we asked to evaluate your manuscript.

Although the Reviewers agree on the potential interest of the manuscript, the issues raised are of a fundamental nature. I will not dwell into much detail, but I would like to highlight the main points.

Reviewer 1 raises a fundamental concern with respect to the actual "fidelity" of the selected Xn-derived tumours with respect to the original and suggests that the evidence provided is insufficient to suggest bona fide human AML tissue. This Reviewer would also like you to provide more convincing arguments for mTORC1 treatment. Reviewer 1 also lists other items of concern that require your action, including inappropriate statistical analysis.

Reviewer 2, similarly to #1, is also concerned about the mechanistic evidence for TGF, PDGF and mTOR as upstream regulators. S/he then focuses on the case for PPARG as a therapeutic target listing a number of issues, and relative actions to be taken, to consolidate your conclusions, including genetic KO of PPARG by CRISPR/CAS, explaining discrepancies in sensitivities to GW, inappropriate in vivo approach, better characterization of the AML model and more. These concerns are of great importance for us as they impinge on the most interesting potential messages of the manuscript.

Finally, I should mention that during our Reviewer cross-commenting exercise, the Reviewers converged and agreed on the need to address all issues.

In conclusion, while publication of the paper cannot be considered at this stage, given the potential interest of your findings and after internal discussion, we have decided to give you the opportunity to address the criticisms.

We are thus prepared to consider a substantially revised submission, with the understanding that the Reviewers' concerns must be addressed with additional experimental data where appropriate and that acceptance of the manuscript will entail a second round of review. The overall aim is to significantly upgrade the relevance and conclusiveness of the dataset, which of course is of paramount importance for our title. As mentioned above, there is a clear need also to improve statistical analysis. This is very close to our hearts at EMBO Press and indeed we ask all authors to take direct action on statistics and other related issues upon revision with a mandatory checklist (see further below). I would also suggest that a more profound and succinctly written introduction and discussion of the evidence supporting the involvement of mTORC1 in AML and the evidence brought forth in this manuscript could balance out the novelty in this manuscript.

I understand that if you do not have the required data available at least in part, to address the above, this might entail a significant amount of time, additional work and experimentation and might be technically challenging, I would therefore understand if you chose to rather seek publication elsewhere at this stage. Should you do so, we would welcome a message to this effect.

***** Reviewer's comments *****

Referee #1 (Comments on Novelty/Model System):

The authors equate the TSc1/2 model in rats with the TSC1/TSC2 situation in humans. However, this is clearly not the case. Evidently this is also why the authors chose to try to explant an AML to try to develop a human system in vitro for better modelling of AML. There are a number of assumptions made in the manuscript that are not necessarily supported: for example why would mTORC1 inhibitors eradicate AML? Indeed AML is a late stage development, so why would mTORC1 treatment have any effect on early stages of the disease? Moreover, the latter was known also from animals models since 2006, where e.g. mTORC1 inhibition resulted in the diminishing of renal cell tumors, but NOT the disappearance of preneoplastic lesions that would regain momentum after mTORC1 inhibition was stopped. Statistics: to use a student's t-test for multiple comparison without a post-hoc test e.g. in Fig 4A is absolutely wrong and in truth embarrassing for the authors.

Referee #1 (Remarks):

The authors are looking at an in vitro/in vivo model for AML to better understand the genesis as well as the treatment of AML. Despite the considerable interest this manuscript could have for the reader of this journal there are some major issues that need to be addressed prior to acceptance of this manuscript for publication.

Introduction: The authors equate the TSc1/2 model in rats with the TSC1/TSC2 situation in humans. However, this is clearly not the case. Indeed, the rodent TSC1/2 symptomatology, pathology and development does not compare easily with the human analogue. Evidently this is also why the authors chose to try to explant AML to try to develop a human system in vitro for better modelling of AML.

There are a number of assumptions made in the manuscript that are not necessarily supported: for example why would mTORC1 inhibitors eradicate AML? Indeed AML is a late stage development, so why would mTORC1 treatment have any effect on early stages of the disease? Moreover, the latter was known also from animals models since 2006, where e.g. mTORC1 inhibition resulted in the diminishing of renal cell tumors, but NOT the disappearance of preneoplastic lesions that would regain momentum after mTORC1 inhibition was stopped.

Materials and Methods /Results

Statistics: to use a student's t-test for multiple comparisons without a post-hoc test (Bonferroni,

Dunnets, Tukey-Kramer etc depending on what the original hypothesis of testing was) e.g. in Fig 4A, is absolutely wrong and in truth embarrassing for the authors.

Although generally employed, the NOD-SCID mice are a severely immune-suppressed "in vivo" system that would not readily represent the "true" situation in a human. Thus in a non-immunoresponding system the re-inoculation of Xn derived cells with ensuing shorter development times represent a selection of malignant cells rather than the selective environment of a complex situation within the confines of a kidney. The question thus can be raised, whether these selected Xn derived tumors/tumor growths have anything to do with the original AML. Indeed, the authors themselves state that the T1 generation has histologically little comparison with an AML tumor whereas later on at T4-XN the lesions had cells representing with blood, lipid and myeloid characteristics. Question however remains whether this could have been achieved with any propagated xenograft or whether this is something specific for the AML xenografts described. The presence of alpha-SMA in T4-XNA, at least to this reviewer, is insufficient proof for a bona-fide human AML tissue, as the authors proposed.

Despite above comments, the authors are to be lauded for the molecular approach aiming to describe the signal transduction pathways expressed in the XN model. Indeed, demonstrating that most markers are present in the XN model and compare well with the known markers in true human AML is a plus. Mechanistic work inhibiting or down-regulating some of the key players in the signal transduction pathway in their Xn model would have added more convincing data. Indeed, focussing on PPAR gamma is certainly interesting as it plays a major role in lipid tissue development. The fact that PPAR gamma expression was not only found in adipocytes is critical to the point that the authors should demonstrate the proportional distribution of endocyte, adipocytes and myelocytes found in T4 or T5-Xn in order to demonstrate that the majority of cells does not consist of adipocytes. Only then could the ensuing experiments be safely interpreted. Indeed, if the majority of cells is truly adipocytes, the effects observed with PPAR gamma inhibition would primarily have to be interpreted as an effect of adipocyte growth inhibition rather than an effect on general PPAR gamma signal transduction inhibition. The latter would also mean that the adipocytes identified still have a certain pluripotency.

Referee #2 (Remarks):

Pleniceanu and colleagues examine the role of PPARG in renal AMLs through in vitro and in vivo models combined with gene expression analysis. They find that PPARG is activated during serial passage generated T4 cells when compared to early passage T1 cells. They also demonstrate that PPARG is activated in human AMLs (by IHC) and that the PPARG inhibitor GW9662 is effective at inhibiting growth in vitro and in vivo although the in vivo experiments are performed in an unconventional manner (see below). Finally, they identify that TGFB1 signaling is downstream of PPARG and suggest that the effects of PPARG are TGFB1 mediated.

The authors should be credited with studying a relatively understudied (although not uncommon) disease. The studies are interesting and the target is clinically relevant. Overall the experiments appear to be well executed. There are some issues that need to be addressed however:

Major Points:

Why do the authors think that the in vivo tumor morphology changed over time?

They appear to show that PPARG signaling is increased in T4/5 tumors but what is the impetus for the increased PPARG signaling? The authors have identified TGFB, PDGF, and mTOR signaling as potential upstream regulators, but are any of these causative?

The PPARG inhibitor GW is not thought to be very specific. To confirm the role of PPARG it seems important to perform parallel experiments using genetic knock-down of CRISPR of PPARG.

Figure 4A. The sensitivity of UMB cells and T4 to GW do not appear to be substantially different. If anything the UMB cells are more sensitive. This seems to be at odds with the overarching hypothesis of the manuscript that T4 cells have enhanced PPARG signaling and therefore PPARG is a therapeutic target in AML. This is a very important point that needs to be resolved.

Figure 4E. Do authors have any thoughts about why the PPARG agonist doesn't seem to increase growth?

Figure 4E should include the same cells as Figure 4D as it is important to evaluate how PPARG agonists affect the same cells.

Figure 6. The in vivo portion of this figure is done quite strangely. The cells are treated in vitro and then implanted into the mouse. But from my understanding, no in vivo treatment is done. To really query whether GW is effective in vivo, established tumors should be treated with in vivo GW. It also seems to be important to look at the histology of treated and untreated tumors. Does PPARG inhibition result in tumors that are less adipogenic, etc?

Figure 7B: I think the authors should check the numbers. I believe that the 1875 and 1987 numbers include some of the overlapping genes.

Why was figure 7 not done with T4 cells since they are thought to be more PPARG dependent?

Since much of the novelty of this paper is related to the development of an in vivo model of AML, better pictures of histology (Figure 1) as well as multiple tumors (supplemental data) need to be shown. It also seems to be important that an expert GU pathologist evaluate the images.

T5 -vs- normal adult kidney (RNA expression) do the differences still hold up? Are the gene expression changes described in Figure 2 just reflect in vitro passaging?

Are there RNA similarities between UMB xenografts versus human AML? The authors already have RNA expression data

Discussion is very long.

Some of the Tables, such as Table 2 could definitely be included as supp tables.

Referee #3 (Comments on Novelty/Model System):

A human AML-xenograft (Xn) model in mice, recapitulating AML at the histological and molecular levels has been established.

Referee #3 (Remarks):

This paper provides a novel concept for treatment of AML. The authors have identified the PPARG pathway as the main signaling system for causing AML. They also provide the first preclinical model to study this otherwise untreatable disease. The experimental data are rigorous and supportive to their conclusions. Appropriate controls are included in each experimental settings. the proposed concept of targeting PPARG for treatment of AML is novel and clinically significant. After minor revision, the manuscript should be accepted for publication.

Minor revision:

The authors should cite the following references:

Sci Adv. 2015 Apr 10;1(3):e1400244.
Nat Commun. 2014 Sep 17;5:4944.
Nat Rev Endocrinol. 2014 Sep;10(9):530-9.
Proc Natl Acad Sci U S A. 2013 Aug 20;110(34):13932-7.
Proc Natl Acad Sci U S A. 2013 Jul 16;110(29):12018-23.
Science. 2013 Jul 5;341(6141):84-7
Trends Mol Med. 2013 Aug;19(8):460-73.
Sci Transl Med. 2011 Dec 21;3(114):114rv3.
Nat Med. 2011 Dec 4;18(1):100-10.

Referee #1 (Comments on Novelty/Model System):

The authors equate the TSc1/2 model in rats with the TSC1/TSC2 situation in humans. However, this is clearly not the case. Evidently this is also why the authors chose to try to explant an AML to try to develop a human system in vitro for better modelling of AML. There are a number of assumptions made in the manuscript that are not necessarily supported: for example why would mTORC1 inhibitors eradicate AML? Indeed AML is a late stage development, so why would mTORC1 treatment have any effect on early stages of the disease? Moreover, the latter was known also from animals models since 2006, where e.g. mTORC1 inhibition resulted in the diminishing of renal cell tumors, but NOT the disappearance of preneoplastic lesions that would regain momentum after mTORC1 inhibition was stopped. Statistics: to use a students t-test for multiple comparison without a post-hoc test e.g. in Fig 4A is absolutely wrong and in truth embarrassing for the authors.

Referee #1 (Remarks):

The authors are looking at an in vitro/in vivo model for AML to better understand the genesis as well as the treatment of AML. Despite the considerable interest this manuscript could have for the reader of this journal there are some major issues that need to be addressed prior to acceptance of this manuscript for publication.

Introduction: The authors equate the TSc1/2 model in rats with the TSC1/TSC2 situation in humans. However, this is clearly not the case. Indeed, the rodent TSC1/2 symptomatology, pathology and development does not compare easily with the human analogue. Evidently this is also why the authors chose to try to explant AML to try to develop a human system in vitro for better modelling of AML.

We thank the referee for this comment. Indeed, the TSC1/2 model in rodents is significantly different compared to the human state, especially with respect to AML, which does not develop in TSC1/2-deficient rodents. Hence, by using human AML-derived cells, we generate for the first time a model of human AML in mice. We have edited the introduction to better emphasize this notion.

There are a number of assumptions made in the manuscript that are not necessarily supported: for example why would mTORC1 inhibitors eradicate AML? Indeed AML is a late stage development, so why would mTORC1 treatment have any effect on early stages of the disease? Moreover, the latter was known also from animal models since 2006, where e.g. mTORC1 inhibition resulted in the diminishing of renal cell tumors, but NOT the disappearance of preneoplastic lesions that would regain momentum after mTORC1 inhibition was stopped.

We thank the referee for this remark. The statement that mTORC1 inhibitors do not eradicate AML was aimed at explaining the necessity to seek additional effective treatments for AML, as well as explain the rationale behind the need to find additional pathways which potentially regulate AML emergence and growth.

Materials and Methods /Results Statistics: to use a students t-test for multiple comparisons without a post-hoc test (Bonferroni, Dunnett, Tukey-Kramer etc depending on what the original hypothesis of testing was) e.g. in Fig 4A, is absolutely wrong and in truth embarrassing for the authors.

We are grateful for this comment. We have reanalyzed the data representing multiple comparisons using one-way analysis of variance (ANOVA) followed by Bonferroni post-hoc tests in figure 4, and added the relevant paragraph in the *materials and methods* section.

Although generally employed, the NOD-SCID mice are an severely immune-suppressed "in vivo" system that would not readily represent the "true" situation in a human. Thus in a nonimmun responding system the re-inoculation of Xn derived cells with ensuing shorter development times represent a selection of malignant cells rather than the selective environment of a complex situation

within the confines of a kidney. The question thus can be raised, whether these selected Xn derived tumors/tumor growths have anything to do with the original AML. Indeed, the authors themselves state that the T1 generation has histologically little comparison with an AML tumor whereas later on at T4-XN the lesions had cells representing with blood, lipid and myeloid characteristics. Question however remains whether this could have been achieved with any propagated xenograft or whether this is something specific for the AML xenografts described. The presence of alpha-SMA in T4-XNA, at least to this reviewer, is insufficient proof for a bona-fide human AML tissue, as the authors proposed.

We thank the referee for this comment. First, to address this issue, we have incorporated in the discussion the limitations of using a model employing immunodeficient mice. Second, our lab has gained much experience in studying xenograft models of human tumors, and we can thus safely determine that the histology of the AML-Xn is not only reflective of the human disease, but also unique in comparison to other tumor Xn models. To demonstrate this concept, we have added figure S2, which shows the histology of Xn models in NOD-SCID mice of two other tumors: Wilms' tumor (WT), representing a renal tumor and Pluero-Pulmonary blastoma (PPB), representing a non-renal tumor. As can be seen in the figure, each tumor has its own unique histology. This is also true for other tumor types studied in our laboratory. Lastly, it should be emphasized that the combined expression of HMB45 and SMA is a unique feature of AML, which when combined with the characteristic histology, is a diagnostic feature of this specific tumor.

Despite above comment, the authors are to be lauded for the molecular approach aiming to describe the signal transduction pathways expressed in the XN model. Indeed, demonstrating that most markers are present in the XN model and compare well with the known markers in true human AML is a plus. Mechanistic work inhibiting or down-regulating some of the key players in the signal transduction pathway in their Xn model would have added more convincing data.

We are grateful for these remarks. As proposed, we indeed show that inhibition of one of the dominant pathways in the Xn model (i.e. the PPARG pathway) results in strong growth-inhibitory effect on AML. To better demonstrate this concept, we have carried out further experiments using another specific PPARG inhibitor, T0007, and shown that AML growth is inhibited using this inhibitor as well. We have also used a molecular approach and demonstrated that PPARG knockdown via shRNA results in growth inhibition of AML cells.

Indeed, focussing on PPAR gamma is certainly interesting as it plays a major role in lipid tissue development. The fact that PPAR gamma expression was not only found in adipocytes is critical to the point that the authors should demonstrate the proportional distribution of endocyte, adipocytes and myelocytes found in T4 or T5-Xn in order to demonstrate that the majority of cells does not consist of adipocytes. Only then could the ensuing experiment be safely interpreted. Indeed, if the majority of cells is truly adipocytes, the effects observed with PPAR gamma inhibition would primarily have to be interpreted as an effect of adipocyte growth inhibition rather than an effect on general PPAR gamma signal transduction inhibition. The latter would also mean that the adipocytes identified still have a certain pluripotency.

We thank the referee for this important comment. Indeed, this issue has been thoroughly examined, and we have detected that only a minority of the cells exhibit an adipocytic phenotype. To emphasize this concept, we have added figure S14, using low magnification, to demonstrate the relatively low proportion of adipocytes in T5 Xn within the complete tumor mass.

In addition, we have addressed and better emphasized this issue in the discussion, where we provide several explanations as to why the upregulation of PPARG is indicative of its role in AML growth rather than adipocytic differentiation:

1 The bioinformatic analysis that we carried out, comparing late and early generation Xn, has demonstrated that various genes regulating tumor growth, non-adipogenic cellular differentiation and the mTOR pathway, are regulated by PPARG, implying that the latter is likely to serve other functions aside from adipogenesis.

2 We show in the paper that PPARG inhibition leads to a robust and specific anti-proliferative effect in AML cells. This indicates that PPARG is a regulator of AML

proliferation. Had the PPARG activation during Xn propagation reflected merely adipogenic differentiation of the cells, we would have expected PPARG inhibitors to block differentiation and reciprocally increase proliferation.

3 Immunostaining for PPARG revealed strong nuclear PPARG expression in both adipogenic and non-adipogenic compartments of AML-Xn, including the large mass of undifferentiated epithelioid cells which are likely to drive tumor growth.

4 A similar expression pattern of PPARG in non-adipogenic tumor components was detected in primary AML. For instance, we show strong PPARG expression in fat-poor AML, indicating that PPARG expression is at least partially independent of adipogenic differentiation in this tumor.

In conclusion, we feel there is sufficient evidence to propose that PPARG expression in AML does not reflect the adipogenic differentiation in this tumor. Rather, we propose that the presence of fat is a by-product of PPARG being a regulator of AML growth.

Referee #2 (Remarks):

Pleniceanu and colleagues examine the role of PPARG in renal AMLs through in vitro and in vivo models combined with gene expression analysis. They find that PPARG is activated during serial passage generated T4 cells when compared to early passage T1 cells. They also demonstrate that PPARG is activated in human AMLs (by IHC) and that the PPARG inhibitor GW9662 is effective at inhibiting growth in vitro and in vivo although the in vivo experiments are performed in an unconventional manner (see below). Finally, they identify that TGFB1 signaling is downstream of PPARG and suggest that the effects of PPARG are TGFB1 mediated.

The authors should be credited with studying a relatively understudied (although not uncommon) disease. The studies are interesting and the target is clinically relevant. Overall the experiments appear to be well executed. There are some issues that need to be addressed however:

Major Points:

Why do the authors think that the in vivo tumor morphology changed over time?

We thank the referee for this important comment. This is an intriguing issue indeed. Relying on our lab's long experience with Xn models of various tumor types, we think that the main factor likely responsible for this change is the continued activity of various differentiation-related pathways on tumor cells within the *in-vivo* environment. Accordingly, several Xn passages are most often required to allow the acquisition of the parental tumor morphology (e.g. differentiation along the 3 lineages in AML). Indeed, in most of our Xn models, including the one presented in this manuscript, the first generation usually exhibits a primitive, disorganized morphology. This may also result from the fact that the Xn originated from a specific, relatively homogenous population of cells (i.e. UMB cells), which requires several *in-vivo* passages to allow differentiation-related processes to take place and give rise to the histology seen at later passages. To address this issue, we have added a short paragraph to the discussion.

They appear to show that PPARG signaling is increased in T4/5 tumors but what is the impetus for the increased PPARG signaling? The authors have identified TGFB, PDGF, and mTOR signaling as potential upstream regulators, but are any of these causative?

We thank the referee for raising this point. Following the demonstration of PPARG activation during *in-vivo* tumor growth and in primary AML specimens, we focused primarily on understanding which downstream effectors link PPARG and AML growth. This was discussed mainly in figure 7 and the related text. We show that PPARG inhibition and AML cell death are accompanied by down-regulation of the TGFB1 pathways, and specifically PDGFb and CTGF. We provide both experimental data and relevant citations that support the link between these factors and perivascular cells, the presumable cell of origin of AML.

The PPARG inhibitor GW is not thought to be very specific. To confirm the role of PPARG it seems important to perform parallel experiments using genetic knock-down of CRISPR of PPARG.

We would like to thank the referee for this remark. In order to better address this issue, we

have carried out and added to the text additional experiments using a different, specific PPARG inhibitor, T0007, and shown that AML growth is inhibited using this inhibitor as well, in a dose-dependent manner. As suggested, we have also used a molecular approach and demonstrated that PPARG knockdown via shRNA results in growth inhibition of AML cells.

Figure 4A. The sensitivity of UMB cells and T4 to GW do not appear to be substantially different. If anything the UMB cells are more sensitive. This seems to be at odds with the overarching hypothesis of the manuscript that T4 cells have enhanced PPARG signaling and therefore PPARG is a therapeutic target in AML. This is a very important point that needs to be resolved.

We thank the referee for this comment. First, to quantify the sensitivity of UMB and T4Xn cells to PPARG inhibition, we calculated the inhibitory concentration 50 (IC50) at 96h treatment. This revealed a slightly higher sensitivity of T4 cells compared to UMB cells, indicating that the former are in fact more sensitive (Figure S4). To further support this notion, we also calculated the IC50 of the subsequent Xn generation, T5, and found that it is even more sensitive than T4 (Figure S4).

This is also reflected in the observation that in the lowest treatment concentration (30mM), T4 cells show highly significant ($p < 0.01$) reduction in cell growth at both 48h and 96h, while UMB cells do not show any significant change at 96h and a smaller, less significant ($p < 0.05$) change at 48h.

In addition, to better assess the relative PPARG-dependence of the different cell types, we carried out and added to the manuscript an experiment whereby the cells were treated with the PPARG *agonist* Rosiglitazone. In the revised manuscript, we show that while UMB and SV7 cells demonstrate a slight, but insignificant increase in cell growth, T5-Xn cells exhibit significant, dose-dependent increase in cell growth in response to PPARG agonist treatment, indicating its PPARG-dependence.

Moreover, it should be emphasized that the direct comparison between UMB cells and T4-Xn cells (or any other Xn cells, for that purpose), inherently includes several important differences and confounding factors. For instance, UMB cells represent a relatively homogenous cell population, constantly grown *in-vitro*, while Xn cells consist of a more heterogeneous population of tumor cells established after *in-vivo* growth. In addition, UMB cells represent an enriched population of tumor-*initiating* cells, while T4-Xn cells represent an enriched population of tumor-*propagating* cells. These are two important, but not necessarily equivalent populations. These differences are one of the main reasons we chose to compare T4-Xn to T1-Xn cells, and not T4-Xn to UMB cells, in the microarray experiment aimed at identifying upregulated pathways during *in-vivo* tumor growth.

Figure 4E. Do authors have any thoughts about why the PPARG agonist doesn't seem to increase growth?

We thank the referee for this remark. As explained in the previous comment, we have revised this issue in the text, to more broadly show the effect of PPARG agonism on all cells types. These results, presented in the revised manuscript, demonstrate that while UMB and SV7 cells show a minimal, insignificant increase in cell growth, T5-Xn cells respond with significant, dose-dependent increase in cell growth in response to PPARG agonist treatment, indicating that they are PPARG-dependent.

Since both UMB and T5-Xn cells are sensitive to PPARG inhibition, this difference in response to PPARG agonism is indeed intriguing.

This difference in response to PPARG agonism may result from several major differences between the UMB cell line the Xn cells.

First, UMB cells represent an *in-vitro* grown population of cells, while Xn cells are derived from the *in-vivo* environment, which could account, at least in part, for the different sensitivity to different stimuli.

Second, UMB cells are a relatively homogenous cell population, possibly expressing PPARG at a level in which further PPARG activation does not result in additional biological effect. In contrast, Xn cells consist of a heterogeneous population of tumor cells, probably consisting of cells with different levels of PPARG expression and/or PPARG-sensitivity, such that at least some of the cells are still prone to respond to PPARG activation with increased proliferation.

And lastly, UMB cells represent an enriched population of tumor-*initiating* cells, while T5-Xn cells represent an enriched population of tumor-*propagating* cells, two important, but not necessarily equivalent populations.

These differences between these two cell types (UMB and Xn cells) are one of the main reasons we chose to compare T4-Xn to T1-Xn, and not to UMB cells, in the microarray experiment aimed at identifying upregulated pathways during *in-vivo* tumor growth.

Figure 4E should include the same cells as Figure 4D as it is important to evaluate how PPARG agonists affect the same cells.

We thank the referee for this comment. We have carried out further experiments using Rosiglitazone on all cell types, and updated the text accordingly. The results are presented in figure 4E, figure S8 and the relevant text. As described above, the results demonstrate that the Xn cells are significantly sensitive to PPARG activation, whereas UMB and SV7 cells show a minimal, insignificant increase in cell growth.

Figure 6. The *in vivo* portion of this figure is done quite strangely. The cells are treated *in vitro* and then implanted into the mouse. But from my understanding, no *in vivo* treatment is done. To really query whether GW is effective *in vivo*, established tumors should be treated with *in vivo* GW. It also seems to be important to look at the histology of treated and untreated tumors. Does PPARG inhibition result in tumors that are less adipogenic, etc?

We thank the referee for this comment. In this part of the manuscript, we have applied a modified version of the tumor initiation/seeding assay, previously used by Gupta et al. 2009, Cell. In this assay, cells are first treated *in-vitro*, and subsequently, only living cells from both the treatment and control groups are injected into mice, in order to test whether the treatment affects their ability to initiate a tumor. Upon treatment with the PPARG inhibitor, a significant portion of Xn cells in the treatment group died (as predicted by the MTS experiments, e.g. Figure 4). Hence, following the treatment, the treatment group is composed of cells that did not express PPARG from the beginning, and were thus unaffected, and cells that expressed PPARG and following treatment express lower PPARG levels and managed to survive nonetheless. Hence, this assay compares AML cells which differ in PPARG expression levels-isolating this feature with respect to the ability to initiate Xn *in-vivo*.

In addition, from the cancer stem cell perspective, the fact that abolishment of PPARG expressing cells resulted in inhibition of tumor initiation, implies that the treatment targets tumor initiating cells, rather than sporadically killing AML cells. This, in turn, suggests that PPARG plays a role in regulating tumor stem cell activity.

And lastly, compounds leading to diminished tumor initiation, as described here for PPARG inhibitor, have been shown to be effective when used as *in-vivo* treatments. The referee is indeed correct as this is not an *in-vivo* treatment experiment, which would be the next step, but out of the scope of this work.

Figure 7B: I think the authors should check the numbers. I believe that the 1875 and 1987 numbers include some of the overlapping genes.

We thank the referee for this comment. Indeed, the referee is correct, we have fixed the figure.

Why was figure 7 not done with T4 cells since they are thought to be more PPARG dependent?

We would like to thank the referee for this question. In this part of the work we chose to use the two original AML cell lines, representing two types of AML tumors, namely, sporadic and TSC-related AML, and therefore potentially capture a more general picture of the effect of anti-PPARG treatment on human AML. In addition, this selection had the added benefit of using more homogenous cell populations, which could lead to more consistent results, with less "noise" related to the inter-sample variability. These cells were also shown to be highly susceptible to PPARG antagonism and therefore were also very suitable for the purpose of this experiment. In addition, we assert that by using different types of cells from the ones which were used to identify PPARG upregulation (i.e. Xn cells), we actually carried out more strict and independent of a test, allowing us to arrive at more reliable results. Nonetheless, in order to test the established results (i.e. downregulation of PDGFB and CTGF following PPARG inhibition), we carried out real-time PCR on T5-Xn treated with PPARG agonist, as suggested by the referee, and demonstrated that the same effect is present in these cells as well. This was added to the text and figure 7.

Since much of the novelty of this paper is related to the development of an in vivo model of AML, better pictures of histology (Figure 1) as well as multiple tumors (supplemental data) need to be shown. It also seems to be important that an expert GU pathologist evaluate the images.

We thank the referee for these important comments. As suggested, we have significantly improved the quality of the pictures. We would like to note that the lower resolution that was used is due to the attempt to save space using smaller files, and not because of lower quality photographing. In addition, we have added a relevant figure to the manuscript (Figure S1), showing two additional tumors, exhibiting the same histological features. Regarding the description of the histological features, it should be noted that the original descriptions, as well as the one provided in the revised manuscript, were given by a certified pathologist specializing in uropathology, which therefore is highly experienced in AML diagnosis.

T5 -vs-normal adult kidney (RNA expression) do the differences still hold up? Are the gene expression changes described in Figure 2 just reflect in vitro passaging? Are there RNA similarities between UMB xenografts versus human AML? The authors already have RNA expression data.

We thank the referee for this suggestion. We have carried out the suggested comparison, between T5-Xn and normal human adult kidney. Indeed, we detected similar differences, namely strong upregulation in AML-Xn (over 21 fold) of *PPARG*, as well as enrichment of some of the most cardinal biological processes characterizing AML. These include angiogenesis, blood vessel morphogenesis, smooth muscle proliferation, muscle differentiation and cellular lipid metabolic processes. We have added the results obtained in this comparison to the main text and figure 2D. As for the comparison between the Xn and primary human AML, this was not feasible as we did not have RNA of primary AML, nor was any relevant bioinformatic data available online at Gene Expression Omnibus (GEO). However, we have addressed the similarity of the AML-Xn to primary human AML with respect to the robust expression of *PPARG*, as demonstrated by immunohistochemical stainings in figure 3.

Discussion is very long.

We thank the referee for this comment. We have revised and shortened the discussion.

Some of the Tables, such as Table 2 could definitely be included as supp tables.

We thank the referee. As suggested, we have moved most tables into the supplementary material section.

Referee #3 (Comments on Novelty/Model System):

A human AML-xenograft (Xn) model in mice, recapitulating AML at the histological and molecular levels has been established.

Referee #3 (Remarks):

This paper provides a novel concept for treatment of AML. The authors have identified the PPARG

pathway as the main signaling system for causing AML. They also provide the first preclinical model to study this otherwise untreatable disease. The experimental data are rigorous and supportive to their conclusions. Appropriate controls are included in each experimental setting. The proposed concept of targeting PPAR γ for treatment of AML is novel and clinically significant. After minor revision, the manuscript should be accepted for publication.

Minor revision:

The authors should cite the following references:

Sci Adv. 2015 Apr 10;1(3):e1400244. *PlGF-induced VEGFR1-dependent vascular remodeling determines opposing antitumor effects and drug resistance to Dll4-Notch inhibitors.*

Nat Commun. 2014 Sep 17;5:4944. *TNFR1 mediates TNF- α -induced tumour lymphangiogenesis and metastasis by modulating VEGF-C-VEGFR3 signalling.*

Nat Rev Endocrinol. 2014 Sep;10(9):530-9. *VEGF-targeted cancer therapeutics-paradoxical effects in endocrine organs.*

Proc Natl Acad Sci U S A. 2013, Aug 20;110(34):13932-7. *Vascular endothelial growth factor-dependent spatiotemporal dual roles of placental growth factor in modulation of angiogenesis and tumor growth.*

Proc Natl Acad Sci U S A. 2013 Jul 16;110(29):12018-23. *Anti-VEGF-and anti-VEGF receptor-induced vascular alteration in mouse healthy tissues.*

Science. 2013 Jul 5;341(6141):84-7 *Monitoring drug target engagement in cells and tissues using the cellular thermal shift assay.*

We thank the referee for his comments and accept his suggestions. All changes have been made as requested.

2nd Editorial Decision

10 January 2017

Thank you for the submission of your revised manuscript to EMBO Molecular Medicine.

We have now received the enclosed reports from the reviewers that were asked to re-assess it. As you will see the reviewers, while generally supportive, do have a number of remaining concerns for you to take action upon.

Providing you deal with the above issues accurately and fully, I am prepared to make an editorial decision on your next, final version

In addition to appropriately addressing the above concerns, please consider the following final editorial amendments:

- 1) Please correct the reference style. If you cannot find the EMBO Molecular Medicine template for your reference manager software, you can use the EMBO Journal one.
- 2) Please provide the "The Paper Explained" section in the manuscript file
- 3) Please include figure legends in main the manuscript text (not as figure captions)
- 4) Callouts for S14 and appendix tables S5, S6, S7 are missing in manuscript. Also, please update callouts to use appropriate appendix nomenclature "Appendix Figure S1", etc. and add a first page TOC to the Appendix.
- 5) Please improve the labelling for all figures as fonts/sizes are not consistent. . Please also note that scale bars are also not consistent or very legible. I suggest you refer to our very useful guide to figure preparation
http://embopress.org/sites/default/files/EMBOPress_Figure_Guidelines_061115.pdf.
- 6) Please indicate area magnified in Figure 1B
- 7) Upon closer inspection with photoshop, we noticed that the Figure 2 panel A has two overlapping labels.

- 8) Please change color of figure 3 labels on micrographs to white to improve readability
- 9) The size bars in figure 8B are difficult to see.
- 10) As per our Author Guidelines, the description of all reported data that includes statistical testing must state the name of the statistical test used to generate error bars and P values, the number (n) of independent experiments underlying each data point (not replicate measures of one sample), and the actual P value for each test (not merely 'significant' or ' $P < 0.05$ ').
- 11) The manuscript must include a statement in the Materials and Methods identifying the institutional and/or licensing committee approving the experiments, including any relevant details (like how many animals were used, of which gender, at what age, which strains, if genetically modified, on which background, housing details, etc). We encourage authors to follow the ARRIVE guidelines for reporting studies involving animals. Please see the EQUATOR website for details: <http://www.equator-network.org/reporting-guidelines/improving-bioscience-research-reporting-the-arrive-guidelines-for-reporting-animal-research/>. Please make sure that ALL the above details are reported in the main text.
- 12) We encourage the publication of source data, with the aim of making primary data more accessible and transparent to the reader. Would you be willing to provide a PDF file per figure that contains the original, uncropped and unprocessed scans of all or at least the key gels used in the manuscript and/or source data sets for relevant graphs? The files should be labeled with the appropriate figure/panel number, and in the case of gels, should have molecular weight markers; further annotation may be useful but is not essential. The files will be published online with the article as supplementary "Source Data" files. If you have any questions regarding this just contact me.
- 13) Every published paper includes a 'Synopsis' to further enhance discoverability. Synopses are displayed on the journal webpage and are freely accessible to all readers. They include a short standfirst as well as 2-5 one sentence bullet points that summarise the paper. Please provide the synopsis including the short list of bullet points that summarise the key NEW findings. The bullet points should be designed to be complementary to the abstract - i.e. not repeat the same text. We encourage inclusion of key acronyms and quantitative information. Please use the passive voice. Please attach this information in a separate file or send them by email, we will incorporate it accordingly. You are also welcome to suggest a striking image or visual abstract to illustrate your article. If you do please provide a jpeg file 550 px-wide x 400-px high.

***** Reviewer's comments *****

Referee #1 (Remarks):

I think the authors tried their best to answer the criticisms raised. Accordingly there remains only little to work on:

Introduction page 3: "This also emphasizes the need.....?"

Although AML was initially considered a hamartoma, it was later shown to be a clonal lesion and thus a true neoplasm (11), prompting the search for its cell of origin. However, the exact identity of the latter has been elusive." The latter sentences do not read well and are not sufficiently logical.

Maybe these can be changed to provide better reading?

There are a number of type-Os that the authors should correct.

Fig 1: As the values are not independent an ANOVA with post-test would be better than a 2-tailed students t test

Fig 4A: The authors show different concentrations of GW9662 and two time-points. The text and the figure legend really do not clearly differentiate what was compared: only the different concentrations and control at 48h, and then the different conc and control at 96h or also the 48h and 96h timepoint? If the former then a one-way ANOVA with Bonferroni correction is OK; whereas in the latter case, a two-way ANOVA would be required again with a post-test. The authors should clarify this in their description and interpretation.

Referee #2 (Remarks):

The authors have made changes to the MS that overall increase the readability and impact. A major issue that still remains the authors response to my comments about Figure 6 is unsatisfactory. As this is a key experiment, I think it needs further scrutiny.

1) Why was a modified version of the tumor initiation/seeding assay by Gupta used? Why not just do the same assay? The modifications the authors use are significantly different enough to consider it a different assay. First, Gupta et al, treated cells for 7 days and then let them recover for 14 days before injecting them into mice. Second, Gupta et al, did a classic TIC assay, using limiting dilution of cells allowing them to calculate the percent of injected cells that have TIC capacity. Finally, at the end of the study, Gupta et al quantified the percent of TICs within the tumors. The assay done by the authors does not reach this level of sophistication.

2) The authors imply that they engraft the same number of the vehicle and GW treated AML-Xn cells and that the cells were "alive". They need to clarify what alive means. Was this just that they were still attached to the plate? That they did not stain positive for trypan blue? That they flow sorted cells that were stained for a viability marker? I would think anything but the latter would be considered.

3) Finally, multiple statements throughout the MS (including the abstract as well as frankly implied in the title) state that PPARG antagonism inhibits *in vivo* growth. I think most cancer biologists would interpret this to mean that an established tumor was treated *in vivo* with an inhibitory compound. I respect the authors point that pretreatment of cells prior to engraftment may demonstrate that PPARG is important for tumor initiating cell capacity.

2nd Revision - authors' response

19 January 2017

Referee #1 (Remarks):

I think the authors tried their best to answer the criticisms raised. Accordingly there remains only little to work on:

Introduction page 3: "This also emphasizes the need.....?"

Although AML was initially considered a hamartoma, it was later shown to be a clonal lesion and thus a true neoplasm (11), prompting the search for its cell of origin. However, the exact identity of the latter has been elusive." The latter sentences do not read well and are not sufficiently logical.

Maybe these can be changed to provide better reading?

We thank the referee for this comment. We have edited these sentences for better reading.

There are a number of type-Os that the authors should correct.

We thank the referee for this remark, we have fixed all type-Os in the text.

Fig 1: As the values are not independent an ANOVA with post-test would be better than a 2-tailed students t test

We have re-analyzed results using one-way ANOVA with post-hoc Bonferroni analysis.

Fig 4A: The authors show different concentrations of GW9662 and two time-points. The text and the figure legend really do not clearly differentiate what was compared: only the different concentrations and control at 48h, and then the different conc and control at 96h or also the 48h and 96h timepoint? If the former then a one-way ANOVA with Bonferroni correction is OK; whereas in the latter case, a two-way ANOVA would be required again with a post-test. The authors should clarify this in their description and interpretation.

We thank the referee for this comment. Indeed the graph represents the comparison between each of the different concentrations and control at 48h, and then the different concentrations and control at 96h, and hence a one-way ANOVA with Bonferroni correction was used. We have edited the text and figure legend to better clarify this issue.

Referee #2 (Remarks):

The authors have made changes to the MS that overall increase the readability and impact. A major issue that still remains the authors response to my comments about Figure 6 is unsatisfactory. As this is a key experiment, I think it needs further scrutiny.

1) Why was a modified version of the tumor initiation/seeding assay by Gupta used? Why not just do the same assay? The modifications the authors use are significantly different enough to consider it a different assay. First, Gupta et al, treated cells for 7 days and then let them recover for 14 days before injecting them into mice. Second, Gupta et al, did a classic TIC assay, using limiting dilution of cells allowing them to calculate the percent of injected cells that have TIC capacity. Finally, at the end of the study, Gupta et al quantified the percent of TICs within the tumors. The assay done by the authors does not reach this level of sophistication.

We thank the referee for this remark. We acknowledge that the assay carried out in our manuscript is less sophisticated than the one used by Gupta et al. Nonetheless, the assay we used is highly relevant to answering the question of whether or not PPARG antagonism is effective in inhibition of tumor initiation, especially as we used a short treatment course of 24h and injected a relatively low cell number of 5×10^4 . Notably, the same assay was carried out in various other papers (e.g. Fan et al. and Bar et al., now referenced in the manuscript). Some of these even used significantly longer ex-vivo pre-treatment before in-vivo injection (up to 7 days). Accordingly, we have omitted the reference to the Gupta article and instead added several references to papers using the same assay as ours for the same purpose.

2) The authors imply that they engraft the same number of the vehicle and GW treated AML-Xn cells and that the cells were "alive". They need to clarify what alive means. Was this just that they were still attached to the plate? That they did not stain positive for trypan blue? That they flow sorted cells that were stained for a viability marker? I would think anything but the latter would be considered.

We thank the referee for this comment. "alive" cells refers to cells that did not stain positive with trypan blue. We have clarified this notion in both the text and figure legend.

3) Finally, multiple statements throughout the MS (including the abstract as well as frankly implied in the title) state that PPARG antagonism inhibits in vivo growth. I think most cancer biologists would interpret this to mean that an established tumor was treated in vivo with an inhibitory compound. I respect the authors point that pretreatment of cells prior to engraftment may demonstrate that PPARG is important for tumor initiating cell capacity.

We thank the referee for this comment. We have modified the title, abstract and manuscript accordingly, to better describe the exact results, emphasizing that PPARG is shown to be important for tumor initiating capacity and that PPARG antagonism was not shown to inhibit *in-vivo* tumor growth.

3rd Editorial Decision

26 January 2017

We are pleased to inform you that your manuscript is accepted for publication and is now being sent to our publisher to be included in the next available issue of EMBO Molecular Medicine.

Corresponding Author Name: Benjamin Dekel

Manuscript Number: EMM-2015-06111-V3